# Taxonomic, molecular and ecological approach reveals high diversity of vector sand flies, varied blood source supply and a high detection rate of *Leishmania* DNA in Colombian Amazon region

Katerine Caviedes-Triana[1,2], Daniela Duque-Granda[1], Gloria Cadavid-Restrepo[1], Claudia X. Moreno-Herrera[1]*, Rafael Vivero-Gomez[1,2]*

1 Grupo de Microbiodiversidad y Bioprospección, Faculty of Sciences, Universidad Nacional de Colombia, Medellín, Colombia, 2 PECET (Programa de Estudio y control de enfermedades tropicales), Facultad de Medicina, Universidad de Antioquia, Medellín, Colombia

* cxmoreno@unal.edu.co (CXM-H); rjviverog@unal.edu.co (RV-G)

## Abstract

### Background

The Amazon region is home to more than 30% of the sand flies species in Colombia, including vectors of *Leishmania* mainly in the genus *Lutzomyia* and *Psychodopygus*. Advances in morphological and molecular taxonomy of sand flies facilitate the development of updated and robust species inventories in understudied areas, such as the departments of Amazonas and Caquetá. Currently, integrating the detection of blood meal sources and *Leishmania* DNA represents a key approach under the "One Health" concept by providing insights into human and animal health and the dynamics of different ecosystems.

### Methodology/principal findings

This study characterized the sand flies fauna in Amazonas and Caquetá using an integrative taxonomic approach that included DNA detection from blood meal and *Leishmania* sources. Sand flies were collected using CDC, Shannon, Prokopack traps and mouth aspirators. DNA was analyzed by conventional PCR targeting COI, Cytb, 12S rDNA and HSP-70N markers, respectively. A total of 1,104 specimens were collected, representing 12 genera and 30 species, 10 are recognized vectors of *Leishmania*, including *Nyssomyia antunesi* and *Psychodopygus amazonensis*. Our findings include new reports of regional distribution, particularly the first report of *Sciopemyia fluviatilis* in Colombia. *Homo sapiens* (28.8% Cytb; 18.6% 12S) and *Sus scrofa* (16.9% Cytb; 6.8% 12S) were the main food sources detected. While *Nyssomyia fraihai* (2.6%), *Trichophoromyia cellulana* (1.3%), *Nyssomyia yuilli pajoti* (1.3%) and *Evandromyia* (*Aldamyia*) *walkeri* (1.0%) grouped the highest detection rate of *Leishmania* DNA (9.0%).

**Data availability statement:** Sequence data associated with the COI marker is available in the GenBank database as follows: PQ783849, PQ783850, PQ783851, PQ783852, PQ783853, PQ783854, PQ783855, PQ783856, PQ783857, PQ783858, PQ783859, PQ783860, PQ783861, PQ783862, PQ783863, PQ783864, PQ783865, PQ783866, PQ783867, PQ783868, PQ783869, PQ783870, PQ783871, PQ783872, PQ783873, PQ783874, PQ783875, PQ783876, PQ783877, PQ783878, PQ783879, PQ783880, PQ783881, PQ783882, PQ783883, PQ783884, PQ783885, PQ783886, PQ783887, PQ783888, PQ783889, PQ783890, PQ783891, PQ783892, PQ783893, PQ783894, PQ783895, PQ783896, PQ783897, PQ783898, PQ783899, PQ783900, PQ783901, PQ783902. Sequence data associated with the Cytb marker is available in the GenBank database as follows: PQ963791, PQ963792, PQ963793 Sequence data associated with the 12S marker is available in the GenBank database as follows: PQ895496, PQ895497, PQ895498, PQ895499, PQ895500.

**Funding:** This study received financial support from project Hermes 57545 of the Universidad Nacional de Colombia, "Alianza estratégica interdisciplinaria Leticia, Medellín y La Paz para el estudio del microbioma de insectos vectores de enfermedades tropicales y su relación con el cambio climático y la sociedad", Convocatoria Nacional para el Fomento de Alianzas Estratégicas interdisciplinarias que articulen los procesos misionales de la Universidad Nacional de Colombia 2022–2024, awarded to CXMH and Scholarship Program of Ministerio de Ciencia, Tecnología e Innovación, Call 15, for Human Capital Development in the context of the Bicentennial and the 2021–2022 Biennial Plan, awarded to KCT. The funders had no role in study design, data collection and analysis, decision to publish, or preparation of the manuscript.

**Competing interests:** The authors have declared that no competing interests exist.

## Conclusions/significance

The integration of molecular tools for the confirmation of phlebotomine species allowed the resolution of taxonomic conflicts, especially in the genus *Trichophoromyia*. These findings provide key information on ecological interactions (vectors-ingesta-*Leishmania*) related to leishmaniasis in the Colombian Amazon, suggesting a high diversity of sand flies and a significant zoonotic potential.

## Author summary

Knowledge of the ecoepidemiology of leishmaniasis, an endemic disease in Colombia, is essential, especially in areas of triple frontier such as the Colombian, Brazil, and Peru Amazon, where the circulation of vectors, hosts and pathogens may be favored. For this reason, we analyzed the ecological interactions of sand flies in areas of transmission of tegumentary leishmaniasis in the departments of Amazonas and Caquetá. Sand flies were identified using an integrative approach combining classical and molecular tools, that also allowed identifying vertebrates from blood remains and detecting *Leishmania* DNA in different environments. Our findings reflect the need to update taxonomic inventories to facilitate tracking of circulating species. We report new records at the departmental and national level and increase the diversity of haplotypes of the COI gene for sand flies. The diversity of vertebrates identified as blood food source of Amazonian sand flies was mainly represented by *Homo sapiens*, *Sus scrofa*, and even monkeys like *Saimiri macrodon*. Demonstrating the importance of understanding the level of dietary plasticity, especially in species vectors, a determining factor in the maintenance of the pathogen life cycle. Finally, the detection of *Leishmania* sp. DNA suggests an active circulation of the parasite in the areas studied.

## 1.  Introduction

Tegumentary Leishmaniases are a group of zoonotic diseases caused by *Leishmania* of the subgenera *Leishmania* and *Viannia*, *L.* (*L.*) *amazonensis*, *L.* (*L.*) *mexicana*, *L.* (*V.*) *braziliensis*, *L.* (*V.*) *guyanensis*, and *L.* (*V.*) *panamensis* in the Colombian Amazon region [1,2]. These parasite species are transmitted by hematophagous Diptera belonging to the subfamily Phlebotominae. A significant number of leishmaniasis cases (4,403) mainly cutaneous (<95%) were confirmed in the Colombian Amazon region alone, of which 37.5% (1,651 cases) were recorded in Amazonas and Caquetá departments during the period of 2020–2024 [3,4].

The fauna and flora biodiversity, the variety of ecologically distinct landscape features, as well as the constant changes in land use in the Amazon region, especially in both departments [5,6], allow the optimal development of sand flies and the increase of pathogen transmission risk [6–8].

In this context, human mobility between forest and jungle areas, driven by tourism, forced displacement, and cross-border economic activities, has led to significant disturbances in natural habitats. Additionally, deforestation, illegal mining, and extensive cattle ranching aggravate the problem of vector-borne diseases (ETVs) such as leishmaniasis [9,10].

This has favored the expansion of the geographical ranges and ecological niches of sand flies [11], increasing their interaction with wild and domestic vertebrates, which play a role in the epidemiological triad as food sources, but also as potential reservoirs of microorganisms [10,12–14].

Recent studies conducted in urban forest remnants and preserved forest environments in the Brazilian Amazon evidenced how the eclectic feeding behavior of sand flies in disturbed habitats increased the risk of contact with vertebrate reservoirs like *Psychodopygus ayrozai* which, besides resenting blood traces of *Homo sapiens*, was also associated with other sources of ingestion such as *Dasypus novemcinctus*, a potential reservoir of *L.* (*V.*) *naiffi* [15].

Other studies have detected blood traces of mammals such as *Caniculus paca* (lowland paca), a potential reservoir of *L.* (*V.*) *lainsoni* [16] and *Tamandua tetradactyla* (collared anteater), proven host of *L.* (*V.*) *guyanensis* [17], in *Nyssomyia antunesi* [16]. *Trypanosoma minasense* DNA has also been detected in individuals of this species along with blood traces of *T. tetradactyla* and *H. sapiens* [15]. Feeding plasticity in these insects favors their survival and establishment in recently colonized habitats or areas where the primary food source is scarce [18,19] as demonstrated in *Ny. umbratilis*, *Lutzomyia gomezi*, and *Ny. antunesi* recorded in Amazonas and Caquetá [20]. Notably, these insects have also been found infected with different *Leishmania* species [21–23].

Therefore, it is essential to understand the extent of dietary plasticity of these insects, especially in areas of the Colombian Amazon, where the availability of vertebrates could influence their role as primary or secondary vectors mainly of cutaneous leishmaniasis (CL) [24,25]. Which contributes to a better understanding of the epidemiology of the disease, particularly in border areas [26].

Specifically, in Belén de los Andaquíes, Milán, Solano and Solita Caquetá, the circulation of 46 species of sand flies distributed in 13 genera has been recorded, with *Psychodopygus* and *Psathyromyia* being the most abundant [20,24,27]. Meanwhile, in Amazonas, studies carried out mainly in the kilometers surrounding Leticia, Amacayacu National Natural Park and Puerto Nariño, have recorded 58 species belonging to 14 genera, the most representative being *Lutzomyia*, *Psathyromyia* and *Psychodopygus* [20,28–30]. These investigations have contributed to the knowledge on the distribution and diversity of sand flies, however, the role of the species in the transmission dynamics of the disease has yet to be addressed.

Entomological studies in the Caquetá department are limited to the year 2000, where species identification was conducted by comparing morphological characters [24]. Although with this approach it is possible to assign species, there are limitations associated with the presence of cryptic species mainly located within the genera *Psychodopygus*, *Nyssomyia* and *Trichophoromyia*. Among the limitations are the collection of individuals of a single sex, isomorphisms in female spermathecae, species complexes, the slow integration of newly described species into taxonomic keys and the use of ambiguous taxonomic characters that limit the correct differentiation of species [29]. For this reason, it is necessary to implement an approach that integrates classical and molecular taxonomy, referred to as "integrative taxonomy", through the use of DNA barcoding and specifically the analysis of partial sequences of the Cytochrome Oxidase Subunit I (COI) gene that serves as a tool for the rapid and accurate identification and delimitation of phlebotomine species at different taxonomic levels and with different degrees of complexity [31–34].

Given the low number of sand flies studies recorded in the Colombian Amazon region and the current relevance of this region, which is highly susceptible to climate change and experiences high circulation of pathogens due to constant human and animal migration, it is crucial to provide new information to better understand the dynamics and risk of leishmaniasis transmission. In this context, the present study characterized the sand fly fauna associated with rural localities in Leticia, Amazonas, and Florencia, Caquetá, using an integrative taxonomy approach (taxonomical keys and COI as barcode), including the detection of *Leishmania* DNA, and the identification of blood ingestion sources through different molecular markers.

## 2. Materials and methods

### 2.1 Collection permits

The collection in the Amazonas and Caquetá departments was conducted under the Framework Permit for Collecting Specimens of Wild Species of Biological Diversity for Non-commercial Scientific Research Purposes, granted by the Autoridad Nacional de Licencias Ambientales (ANLA) to the Universidad Nacional de Colombia by Resolution No. 0255 of March 14, 2014 (article 3), in accordance with the Decree No. 1376, 2013 of the Ministerio de Ambiente y Desarrollo Sostenible, considering the mobility certificate 59453 and with prior consent of the landowners and indigenous community.

### 2.2 Study area

An entomological survey was conducted in San Pedro de los Lagos (S: -4.142304, W: -69.953815) and Tanimboca (S: -4.127948276, W: -69.95326736) localities of Leticia, Amazonas, and in Jericó (N:1.72855556, W: -75.64913), Sebastopol (N: 1.65894444, W: -75.605111), Santo Domingo (N: 1.59536111, W: -75.657417) localities and in the Amazonian Research Center-CIMAZ-Macagual (N: 1.61750000, W: -75.616750) of Florencia, Caquetá (Fig 1).

The study area is part of the Colombian Amazon region, characterized by the presence of tropical rainforest (Bh-T) [35]. The average annual temperature in the department of Amazonas ranges between 25.3°C and 25.7°C, the annual rainfall varies between 2,660 mm and 3,538 mm and the relative humidity is above 80% [36]. In the department of Caquetá, the average temperature ranges from 24.8°C to 25.9°C, the annual average precipitation varies between 4,385 mm and 2,483 mm and relative humidity is above 80% [36]. The most important productive activities in the region are livestock, agriculture, forestry, mining, ornamental fishing, and border trade [9,37].

Sand flies collection was conducted in secondary and primary forests. In the department of Amazonas, near the collection area, cassava and sugarcane crops, palm and fruit trees, domestic animals such as *Canis lupus* (dogs), *Felis catus*

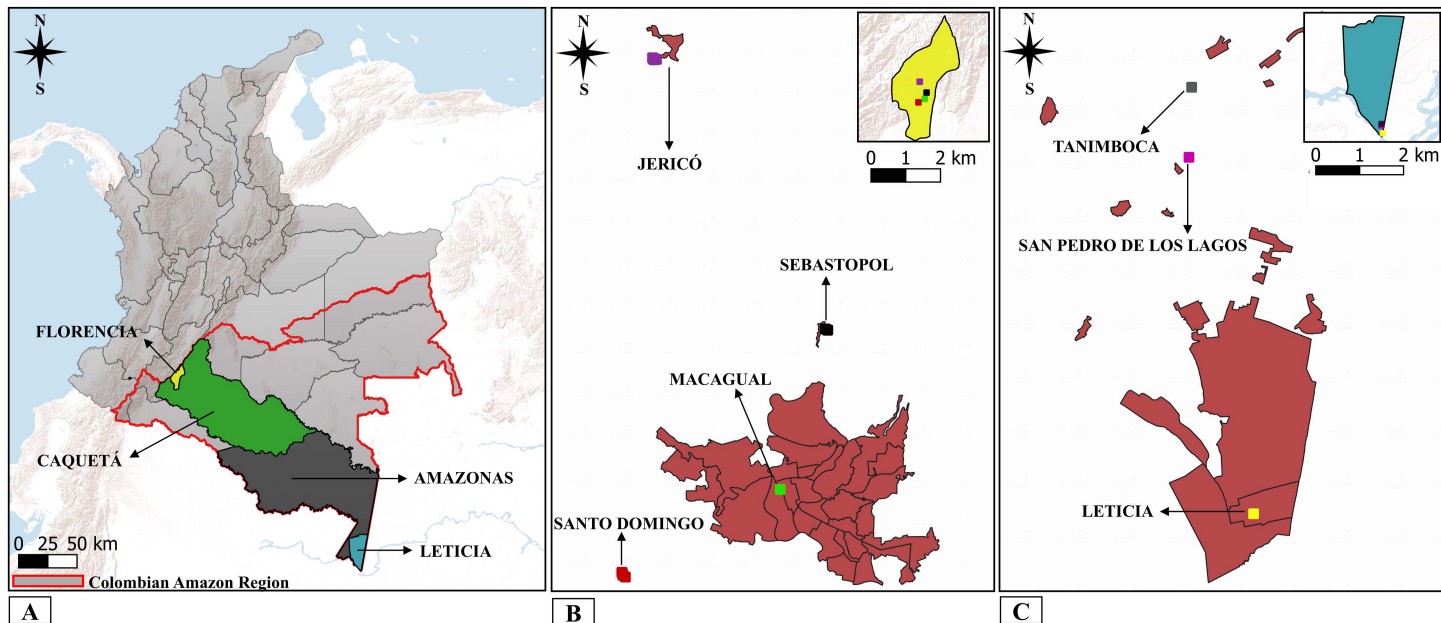

**Fig 1. Geographic location of the study area and sand flies collection sites: (A) Departments of Amazonas and Caquetá, Colombia.** (B) Municipality of Florencia, locality Jericó, Macagual, Santo Domingo and Sebastopol. (C) Municipality of Leticia, locality Leticia, San Pedro de los Lagos and Tanimboca. Source of layers [39–42] and Qgis, v. 3.34.12 [43].

(cats) and *Gallus gallus* (hens) were observed. In the department of Caquetá, traps were placed in areas near vegetable gardens, arazá crops, cocoa, coffee, and banana plantations, all located close to natural water sources, and with presence of domestic animals such as *Sus scrofa* (domestic pig), *C. lupus*, *F. catus*, *G. gallus*, and *Oryctolagus cuniculus* (rabbits). However, in both departments, no evidence of animals associated with livestock was observed.

Sampling sites selection was made considering access conditions to the area, the history of leishmaniasis cases reported by health departments [38] and the existence of favorable microenvironments for sand flies development.

### 2.3 Sand flies collection and processing

During the year 2023, two collection campaigns were carried out, the first in August in Caquetá and the second in November in Amazonas. In each locality of both departments, 12 CDC (Centers for Disease Control and Prevention) light traps were installed in intra, peri and extradomiciliary areas, where they operated for 12 hours (18:00–6:00) each night, for three consecutive days. On the second night of collection, a Shannon trap was set up for one hour (18:00–19:00) at the point where the CDC traps recorded the highest abundance of sand flies on the first night. The collection was complemented with a search in potential resting sites, such as walls chicken coops, rabbit hutches and pig housing, using mouth and Prockopack [44] aspirators up for one hour (18:00–19:00). Each collection point was georeferenced with a GPS-map_60CSx, GARMIN. Specimens were stored and transported dry in 1.5 ml tubes, in racks with silica gel, and then to the laboratory of medical entomology of the Program for the Study and Control of Tropical Diseases (PECET), from the Universidad de Antioquia where all samples were stored at -20°C until processing.

Collected insects were fragmented for taxonomic identification. The head and the last three abdominal segments were mounted in Euparal medium (Carl Roth GmbH + Co. KG, Germany) for taxonomic allocation using the Galati classification [45], supported by the key of Young and Duncan [46] and bibliographical records of sand flies in the Colombian and Brazilian Amazon region. Some identified specimens were deposited in the Francisco Luis Gallego - MEFLG Entomological Museum of the Universidad Nacional de Colombia, at Medellín. Codes: 5089–65091, 65079–65081, 65082–65088, 65142–65146, 65147–65156, 65160–65162, 65157–65159, 65141, 65078. The use of generic and subgeneric abbreviations was conducted following Marcondes [47].

### 2.4 Molecular analyses: DNA extraction of sand flies

After sand flies identification by classical taxonomy, DNA extraction was performed from the thorax, legs, wings, and proximal abdominal segments of specimens using a high salt concentration protocol [48]. Individual abdomens from fed females were differentiated, separated and preserved in RNAlater until DNA extraction and blood source identification. Previous to extraction and in order to separate the tissue from the preservative salts, each specimen was transferred under sterile conditions to a new vial and washed with 70% ethanol for one minute at 2.000 revolutions per minute (rpm), then transferred to a new vial and washed with ultrapure water for one minute at 2.000 rpm, for finally transferred to the vial where DNA extraction would be performed.

The extraction process included mechanical lysis with pistil, thermal shock lysis, protein precipitation with proteinase K (Scientific Inc., Waltham, MA, USA), washing with alcohol at 70 and 96%, and final elution in TE Buffer (1X). DNA concentrations were determined using a NanoDrop spectrophotometer (IMPLEN, Germany) and stored at –20°C for molecular identification, blood source and *Leishmania* detection.

**2.4.1 PCR for phlebotomine barcoding.** The representation and total DNA of four females and four males of each phlebotomine species previously identified by classical taxonomy was considered, using partial amplification of the COI gene, using the primers LCO1490 5'-GGTCAACAAATCATAAAGATATTGG-3' and HCO2198 5'-TAAACTTCA GGGTGACCAAAAAATC A-3' [49], amplifying a product of approximately 710 base pairs (bps). PCR reaction contained 1X buffer, 0.25 mM $MgCl_2$, 0.2 mM dNTPs, 0.4 µM of each oligonucleotide (forward and reverse), 1.5 U of Taq DNA polymerase (Excel taq, SMOBIO Technology, Hsinchu Science Park, Taiwan), 1mg/ml BSA (Scientific Inc., Waltham, MA,

USA), 10–25 ng/µl of total DNA and ultrapure water until a final reaction volume of 25 µl was completed [31]. *Ny. antunesi* DNA was previously used in PCR standardization, whose identity was confirmed by sequencing, and which was negative for *Leishmania* infection and blood ingestion, was included positive control and ultrapure water as a negative control. The thermal profile consisted of an initial cycle of 1 min at 94°C, 1 min at 58°C and 2 min at 72°C followed by 35 cycles of 15 s at 94°C, 1 min at 58°C, 1.30 min at 72°C and finally a cycle of 15 s at 94°C, 1 min at 58°C, 7 min at 72°C [31].

**2.4.2 Blood-meal source identification.** To increase the probability of detecting host DNA at different levels of preservation within the gut of females identified with blood traces, two molecular markers were used in independent PCR reactions. In the first reaction, the partial region of the Cytochrome B gene (Cytb) was amplified using the primers L14841 5'-AAAAAGCTTCCATCAACATC-3' and H15148 5'-AAACTGCAGCCCCTCAGGATTTTGTTCCTC-3' that amplify a fragment of approximately 305 bps [50]. In the second reaction, the 12S ribosomal gene was targeted with the primers L1085 5'-CCCAAACTGGGATTAGATACCC-3' and H1259 5'-GTTTGCTGAAGATGGCGGTA-3' amplifying a fragment of approximately 215 bps [51]. PCR reaction contained 1X buffer, 1.5 mM MgCl2, 0.2 mM dNTPs, 0.5 µM of each oligonucleotide (forward and reverse), 1.5 U of Taq DNA polymerase (Excel taq, SMOBIO Technology, Hsinchu Science Park, Taiwan) 1 mg/ml BSA (Scientific Inc., Waltham, MA, USA), 10–25 ng/µl of total DNA and ultrapure water until a final reaction volume of 25 µl was completed. The thermal cycling conditions consisted of an initial cycle of 5 min at 95°C, 35 cycles of 30 s at 95°C, 15 s at 57°C, 30 s at 72°C and finally a 10 min cycle at 72°C [52]. DNA from *Aedes aegypti* fed with *H. sapiens* and *C. lupus* was used as a positive control and *Ae. aegypti* Rockefeller strain without blood meal ingestion, donated by the medical entomology laboratory of PECET, was also included as a negative control.

**2.4.3 Molecular detection of *Leishmania* DNA.** The N-terminal region of the gene encoding the heat shock protein 70 (HSP-70) was amplified from the DNA of 37.9% of all females collected (with and without evidence of blood ingestion) to detect the presence of *Leishmania* DNA. These specimens were then organized into 168 samples, of which 41 contained 183 females grouped in pools of maximum six specimens, the remaining 127 samples belonged to individual females.

The protocol described by Montalvo *et al*. was used, with the primers F25 5'-GGACGCCGGCACGATTKCT-3' and R617 5'-GAAGAAGTCCGATACGAGGGA-3' amplifying a product of approximately 593 bps [53], *L.* (*V.*) *braziliensis* strain WA140 was donated by PECET and was used as a positive control, and *Ny. antunesi* DNA of wild type previously negative for parasite detection was used as a negative control.

PCR reaction contained 1X buffer, 1 mM MgCl$_2$, 0.2 mM dNTPs, 0.4 µM of each oligonucleotide (forward and reverse), 1 U of Taq DNA polymerase (Excel taq, SMOBIO Technology, Hsinchu Science Park, Taiwan), 1 mg/ml BSA (Scientific Inc., Waltham, MA, USA), 10–40 ng/ µl of total DNA and ultrapure water until a final reaction volume of 25 µl was completed. The thermal cycling conditions consisted of an initial cycle of 1 min at 95°C, 35 cycles of 30 s at 95°C, 45 s at 61°C, 45 s at 72°C and finally a 10 min cycle at 72°C [53].

## 2.5 Sequence analysis PCR products

The PCR fragments were visualized with the EZ-Vision dye (Amresco, USA) on an agarose gel at 1.5%. 100 bp plus (Scientific Inc., Waltham, MA, USA) molecular weight marker was used, and the images were captured through the transilluminator Essential V6 (UVITEC-CAMBRIDGE). PCR products of the expected size were subjected to DNA sequencing in both directions using ABI PRISM 3700 DNA analyzer service of Applied Biosystem.

COI, 12S rRNA and Cytb sequences were edited and aligned using the Finch TV software V 4.0 (Geospiza, Inc.) and MEGA (Molecular Evolutionary Genetics Analysis) programs respectively [54] X v11. Sequence similarity analysis was performed by comparing sequences recorded in the GenBank database with the Basic Local Alignment Tool (BLASTN) [55].

The filtered reference sequences were downloaded, and multiple alignments were constructed with the sequences from this study using ClustalW with predetermined parameters. The phylogenetic analysis was performed by constructing

dendrograms with the neighbor-joining (NJ) method with genetic distances per pair and a bootstrap value of 1000 replicates in the MEGA v11 software. Specifically in the COI marker sequences, the pair-based genetic distances for the maximum intra-specific and minimum inter-specific distances (nearest neighbor, NN) were generated using the Analysis tool with uncorrected distances (p) and the Kimura 2-parameter model (K2P) in the MEGA v11 software, while the number of haplotypes (h), polymorphic sites (s), and nucleotide diversity were determined using DnaSP 6.12.03 [56].

The graphs were made in the Linux environment. The sequence of *Nemopalpus* sp. (KM896679.1) was used as the external group for the COI dendrogram, while *Bufo montfontamus* (MK284968.1) was the external group selected for the blood intake dendrogram.

All sequences obtained in this study were submitted to GenBank with the following access codes: Cytochrome Oxidase Subunit I (COI): PQ783849 - PQ783902. Cytochrome B gene (Cytb): PQ963791 - PQ963793. Ribosomal 12S gene (12S): PQ895496 - PQ895500.

### 2.6 Analysis of interactions between blood-meal sources, *Leishmania* and sand flies species

The standardization of ranges in relation to the abundance of each species for each study site was calculated by the Standardized Index of Species Abundance (SISA), described by Roberts and Hsi [57]. The distribution and diversity of species was analyzed using the ANAFAU software, following the methodology described by Silveria *et al*. [58]. Through this analysis, species were classified into three categories: constant (species present in more than 50% of catches), incidental (present in 25–50% of catches) or accidental (present in less than 25% of catches). Shannon's Diversity (H) and Equity (J) indices were used to estimate species richness and abundance uniformity. The alpha diversity indices were calculated in the PAST software (V 4.10). Diversity was calculated by the Simpson 1-D index, dominance by (Dominance_D) and equity was calculated by the Shannon_H index. The distribution of species was performed by a Principal Component Analysis (PCoA) with Bray-Curtis distances.

To estimate the infection rate in positive *Leishmania* samples, the minimum infection rate is the following formula: Minimal Infection Rate (MIR) = Number of infected sand flies/Total sand flies tested x 100. Finally, circus plot plots were used to show the interactions between sand flies species, blood intake source, *Leishmania* and their detection frequencies.

## 3. Results

### 3.1 Sand flies fauna recorded in the departments of Amazonas and Caquetá

A total of 1,104 sand flies were collected, with females comprising 68.4% (N = 755) and males 31.6% (N = 349). Of these, 191 specimens (17.3%) were collected in Amazonas and 913 (82.7%) in Caquetá. The specimens belonged to 30 species distributed across 12 genera. The most species-rich genera were *Psychodopygus* (7 species), *Nyssomyia* (4 species), *Trichophoromyia* (3 species) and *Evandromyia* (3 species) (Table 1).

When considering the relative abundances between the two departments, *Nyssomyia* (n = 449, 40.7%), *Trichophoromyia* (n = 252, 22.8%), *Evandromyia* (n = 136, 12.3%) and *Psychodophygus* (n = 133, 12.0%) represented the most abundant genera (Fig 2), while at species level *Th*. *cellulana* (n = 236, 21.4%), *Ny*. *yuilli pajoti* (n = 205, 18.5%), *Ev*. (*Ald*.) *walkeri* (n = 133, 12.0%), *Ny*. *antunesi* (n = 122, 11.0%) and *Ny*. *fraihai* (n = 120, 10.8%) were the most abundant. Specifically in Amazon, *Ev*. (*Ald*.) *walkeri* (n = 45, 23.6%) and *Ps*. *amazonensis* (n = 26, 13.6%) were the most abundant species, while in Caquetá it was *Th*. *cellulana* (n = 236, 25.9%), *Ny*. *yuilli pajoti* (n = 205, 22.5%) and *Ny*. *antunesi* (n = 120, 13.3%).

### 3.2 Ecological descriptors of sand flies captured

The analysis of the standardized index of species abundance (SISA) by sampling area shows that *Ny*. *umbratilis* (SISA = 1; 28.6%) was the most abundant species associated with the Tanimboca locality, while in San Pedro de los Lagos *Ev*. (*Ald*.) *walkeri* (SISA = 0.4; 24.4%) and *Ps*. *amazonensis* (SISA = 0.9; 13.5%) were the most abundant species.

**Table 1. Taxonomic inventory of sand flies, sex and feeding status of the females collected in the departments of Amazonas (Tanimboca and San Pedro de los Lagos localities) and Caquetá (Sebastopol, Jericó, Macagual, and Santo Domingo localities).**

| Department | | Caquetá | | | | | | | | San Martín | | | Amazonas | | | | | N (%) | |
|---|---|---|---|---|---|---|---|---|---|---|---|---|---|---|---|---|---|---|---|
| Municipality | | Caraño | | | | | Santo Domingo | | | | | | Leticia | | | | | | |
| Locality | | Sebastopol | | Jericó | | | Santo Domingo | | | Macagual | | | Tanimboca | | San Pedro de los Lagos | | | | |
| Genus | Species | ♀ | ♂ | ♀ | ♀ˆ | ♂ | ♀ | ♀ˆ | ♂ | ♀ | ♀ˆ | ♂ | ♀ | ♀ˆ | ♀ | ♀ˆ | ♂ | | |
| *Brumptomyia* | 1 *mesai* ** | - | - | - | - | 1 | - | - | - | - | - | - | - | - | - | - | - | 1 | (0.1) |
| *Evandromyia* | 2 *(Eva.) georgii* ** | - | - | - | 2 | - | - | - | - | - | - | - | - | - | - | - | - | 2 | (0.2) |
| | 3 *(Eva.) saulensis* | - | - | - | - | - | - | - | - | - | - | - | 1 | - | - | - | - | 1 | (0.1) |
| | 4 *(Ald.) walkeri* | - | - | 4 | 9 | 73 | - | - | 1 | 1 | - | - | - | - | 17 | 7 | 21 | 133 | (12.1) |
| *Nyssomyia* | 5 *antunesi* + | - | - | - | - | - | 4 | 1 | 13 | 101 | 1 | - | - | - | 1 | - | 1 | 122 | (11.1) |
| | 6 *fraihai* ** | 8 | - | 28 | 4 | - | 40 | 1 | 7 | 17 | - | - | - | - | 14 | 1 | - | 120 | (10.9) |
| | 7 *umbratilis* + | - | - | - | - | - | - | - | - | - | - | - | 1 | 1 | - | - | - | 2 | (0.2) |
| | 8 *yuilli pajoti* ** | 1 | - | - | 2 | 1 | 17 | 8 | 8 | 166 | 1 | 1 | - | - | - | - | - | 205 | (18.6) |
| *Lutzomyia* | 9 *(Tri.) sherlocki* ** | - | - | 1 | - | - | - | - | 1 | - | - | - | - | - | - | - | 1 | 3 | (0.3) |
| | 10 *(Hel.) tortura* ** + | - | - | - | - | - | 25 | - | 15 | - | - | - | - | - | - | - | - | 40 | (3.6) |
| *Micropygomyia* | 11 *(Mic.) pilosa* | - | - | - | - | - | - | - | 1 | - | - | - | - | - | - | - | - | 1 | (0.1) |
| *Psychodopygus* | 12 *amazonensis* + | - | 2 | - | - | - | - | - | - | - | - | - | 1 | - | 22 | - | 3 | 28 | (2.5) |
| | 13 *ayrozai* + | - | - | - | - | - | 2 | 1 | 2 | - | - | - | - | 1 | 4 | - | 5 | 15 | (1.4) |
| | 14 *carrerai thula* ** | - | - | - | - | - | 3 | - | 2 | - | - | - | - | - | - | - | - | 5 | (0.5) |
| | 15 *chagasi* + | 11 | 13 | - | - | - | 2 | - | 5 | - | - | - | - | - | - | - | - | 31 | (2.8) |
| | 16 *davisi* + | - | - | - | - | - | 5 | - | - | - | - | - | - | - | 12 | 1 | 6 | 24 | (2.2) |
| | 17 *paraensis* + | - | - | - | - | 1 | 6 | - | 3 | - | - | - | - | - | 15 | 2 | 2 | 29 | (2.6) |
| | 18 *panamensis* + | - | - | 1 | - | - | - | - | - | - | - | - | - | - | - | - | - | 1 | (0.1) |
| *Psathyromyia* | 19 *(For.) aragaoi* | 4 | 6 | - | - | - | - | - | - | - | - | - | - | - | 3 | - | 3 | 16 | (1.5) |
| | 20 *(Psa.) dendrophyla* | - | - | - | - | - | - | - | 1 | - | - | - | - | - | - | - | - | 1 | (0.1) |
| *Pintomyia* | 21 *(Pif.) serrana* ** | - | - | - | - | - | 1 | - | 1 | - | - | - | - | - | - | - | - | 2 | (0.2) |
| | 22 *(Pif.) nevesi* | - | - | - | - | - | 1 | - | - | - | - | - | - | - | - | - | - | 1 | (0.1) |
| *Sciopemyia* | 23 *fluviatilis* * ° | - | - | - | - | - | - | - | - | - | - | - | 1 | - | 2 | - | - | 3 | (0.3) |
| | 24 *sordellii* | - | - | - | - | - | - | - | - | - | - | - | 1 | 1 | - | - | - | 2 | (0.2) |
| *Trichophoromyia* | 25 *cellulana* | - | - | 112 | 8 | 116 | - | - | - | - | - | - | - | - | - | - | - | 236 | (21.4) |
| | 26 *howardi* ** | - | - | - | - | 2 | - | - | - | - | - | - | - | - | - | 5 | 7 | 14 | (1.3) |
| | 27 *velezbernali* | - | - | - | - | - | - | - | - | - | - | - | - | - | 2 | - | - | 2 | (0.2) |
| *Trichopygomyia* | 28 *witoto* | - | - | - | - | - | - | - | - | - | - | - | - | - | - | - | 15 | 15 | (1.4) |

*(Continued)*

**Table 1.** (Continued)

| Department | | Caquetá | | | | | | | | | | Amazonas | | | | | N (%) | |
|---|---|---|---|---|---|---|---|---|---|---|---|---|---|---|---|---|---|---|
| Municipality | | Caraño | | | | | Santo Domingo | | San Martín | | | Leticia | | | | | | |
| Locality | | Sebastopol | | Jericó | | | Santo Domingo | | Macagual | | | Tanimboca | | San Pedro de los Lagos | | | | |
| Genus | Species | ♀ | ♂ | ♀ | ♀ˆ | ♂ | ♀ | ♀ˆ | ♂ | ♀ | ♀ˆ | ♂ | ♀ | ♀ˆ | ♀ | ♀ˆ | ♂ | | |
| *Viannamyia* | 29 *caprina* * | - | - | - | - | - | - | - | - | - | - | - | 1 | - | - | - | - | 1 | (0.1) |
| | 30 *tuberculata* *+ | - | - | - | - | - | - | - | - | - | - | - | - | - | 1 | - | - | 1 | (0.1) |
| *Lutzomyia* | 31 *sp.* | 3 | - | 11 | 1 | 3 | 8 | - | 3 | 8 | 1 | - | 1 | - | 5 | - | 3 | 47 | (4.3) |
| **Total for sex** | | 27 | 21 | 157 | 26 | 197 | 114 | 11 | 63 | 293 | 3 | 1 | 5 | 2 | 100 | 17 | 67 | 1104 | |
| **Total for locality** | | 48 | | 380 | | | 188 | | 297 | | | 7 | | 184 | | | | |
| **Total for department** | | 913 | | | | | | | | | | 191 | | | | | | |

*New record for the department of Amazonas, ** New record for the department of Caquetá, ° New record for Colombia, + Species of epidemiological relevance, **N (%)**: Relative abundance of captured sand flies, ♀: Female, ♀ˆ : Fed females, ♂: Male.

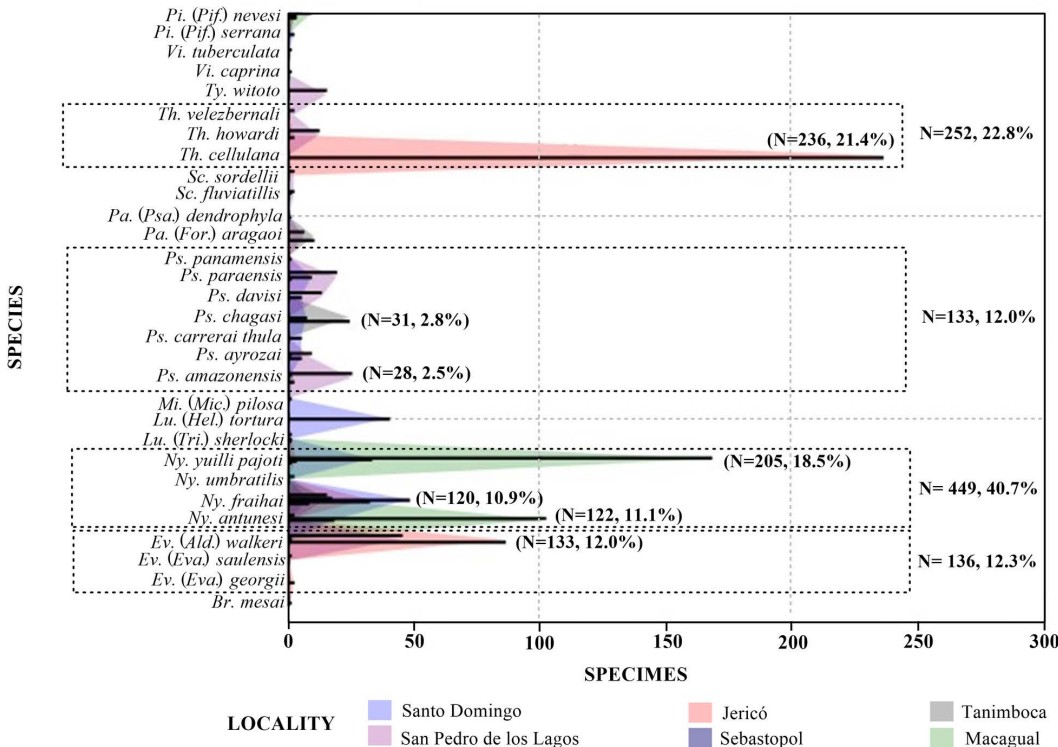

**Fig 2. Relative abundances of sand flies species collected in the departments of Amazonas (Tanimboca and San Pedro de los Lagos) and Caquetá (Sebastopol, Jericó, Macagual and Santo Domingo). Dotted line**: most abundant genera, **N**: Total specimens by species and by genus, **%**: Percentage abundance by species and by genus.

Conversely, *Ev.* (*Eva.*) *saulensis*, *Lu.* (*Trl.*) *sherlocki*, *Ny. antunesi*, *Sciopemyia fluviatilis*, *Sc. sordellii*, *Viannamyia tuberculata*, and *Th. velezbernali* were classified as accidental species (Chao-1 = 16.6) associated with the department of Amazonas (S1 Table). Regarding Caquetá, the most abundant species in Jericó was *Th. cellulana* (SISA = 1; 62.1%),

followed by *Ev.* (*Ald.*) *walkeri* (SISA = 0.88; 22.6%). In Sebastopol, *Ps. chagasi* (SISA = 0.96; 55.3%) and *Pa.* (*For.*) *aragaoi* (SISA = 0.85; 20.8%) predominated.

Ny. *fraihai* (SISA = 0.7; 25.5%) was the most abundant in Santo Domingo, followed by *Lu.* (*Hel.*) *tortura* (SISA = 1; 21.3%) while *Ny. yuilli pajoti* (SISA = 0.98; 56.5%) and *Ny. antunesi* (SISA = 0.98; 34.3%) were predominated in San Martín; *Ev.* (*Ald.*) *walkeri*, *Lu.* (*Trl.*) *sherlocki*, *Micropygomyia* (*Mic.*) *pilosa*, *Ps. davisi*, *Pa.* (*Psa.*) *dendrophyla* and *Ps. paraensis* were classified as accidental and sporadic species (Chao-1 = 20) (S1 Table).

Alpha diversity indices evidenced that species richness varied considerably between locations. Santo Domingo, and San Pedro de los Lagos exhibited the highest richness with 15 and 16 species respectively, represent the localities with the greatest diversity (Simpson 1-D: 0.8235 and 0.8676), lowest dominance (Dominance_D: 0.1765 and 0.1324), and have an equitable distribution of species (Shannon_H: 2.014 and 2.268). Jericó and Macagual presented the highest dominance values (Dominance_D of 0.4814 and 0.4692), the lowest diversity indices (Simpson 1-D: 0.5186 and 0.5308) and the lowest evenness (Shannon_H: 0.9971 and 0.8676) (S1 Fig).

The PcoA analysis with Bray-Curtis distances exhibits clear differences in species composition between localities (Fig 3). In plane I species are grouped based on ecological indices of low dominance and abundance (D: ND-D; A:C-MA). These species are similar in terms of their distribution and abundance, indicating a balance in the community. Plan II mainly represents the most abundant and dominant species. In some sites, one or more species predominated within the community, such as *Th. cellulana* in Jericó Caquetá, which could influence community dynamics, affecting resource availability and interactions among species.

Conversely, planes III and IV include non-dominant species, characterized by their rarity and low specimen counts. Their low frequency may be associated with unfavorable environmental conditions or limited interactions with dominant species, which could contribute to their displacement.

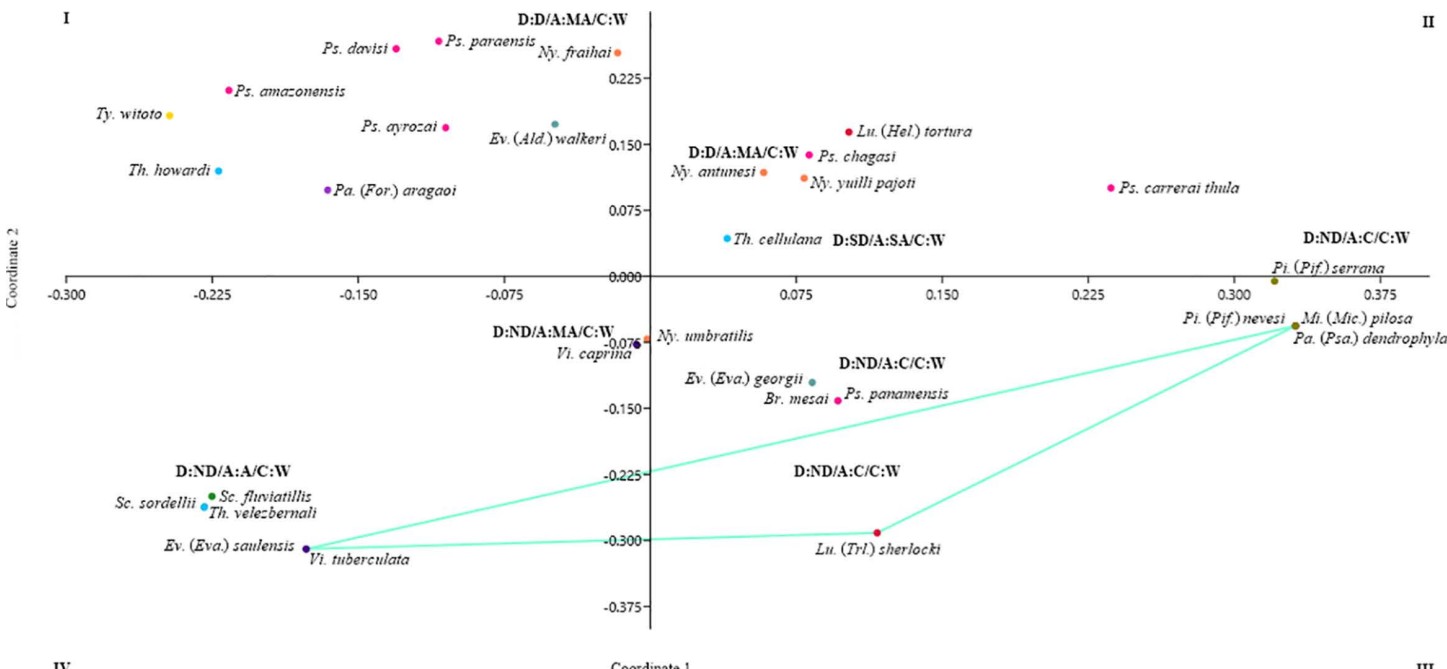

**Fig 3. Principal component analysis (PCoA) indicates the distribution of sand flies species from Bray-Curtis similarity and standardized species abundance index (SISA). SD**: super dominant, **D**: dominant, **ND**: non-dominant; **SA**: super abundant, **MA**: Very abundant, **C**: common, **A**: Accidental, **W**: Constant.

### 3.3 Sequence analysis, genetic diversity, and phylogenetic information

A total of 97 sand flies sequences were obtained, representing 12 genera and 30 species from the surveyed localities in the Colombian Amazon. For most species, between three and nine specimens were included in the analysis. However, in cases of low abundance, only one or two specimens were analyzed. Additionally, 16 new COI sequences were obtained for the species *Lu.* (*Hel.*) *tortura* (6), *Th. cellulana* (6), *Trichopygomyia witoto* (3) and *Vi. caprina* (1).

After manually editing, the consensus sequences fluctuated from 550 to 675 bps. The fragment was aligned between positions 17 and 669 of the 5' segment of the mitochondrial COI gene of *Ae*. *aegypti* (OM214532.1) used as a reference genome, a region that coincides with that proposed by Hebert *et al*. as a barcode for species identification [59]. Visual inspection of alignment indicated the absence of termination codons in the medium of sequences suggesting the presence of pseudogenes or nuclear copies of mitochondrial origin NUMTs. The similarity percentage of between the sequences obtained when compared with the COI sequences of neotropical sand flies available at GenBank ranged from 93 to 100%.

During the nucleotide alignment including only the sand flies identified in our study, 442 preserved sites and 134 polymorphic sites were identified, of which 57 are sparingly informative. This is reflected by the nucleotide heterogeneity according to FINGERPRINT in Fig 4A. The A+T ratio (66.7%) was significantly higher than the G-C ratio (33.3%).

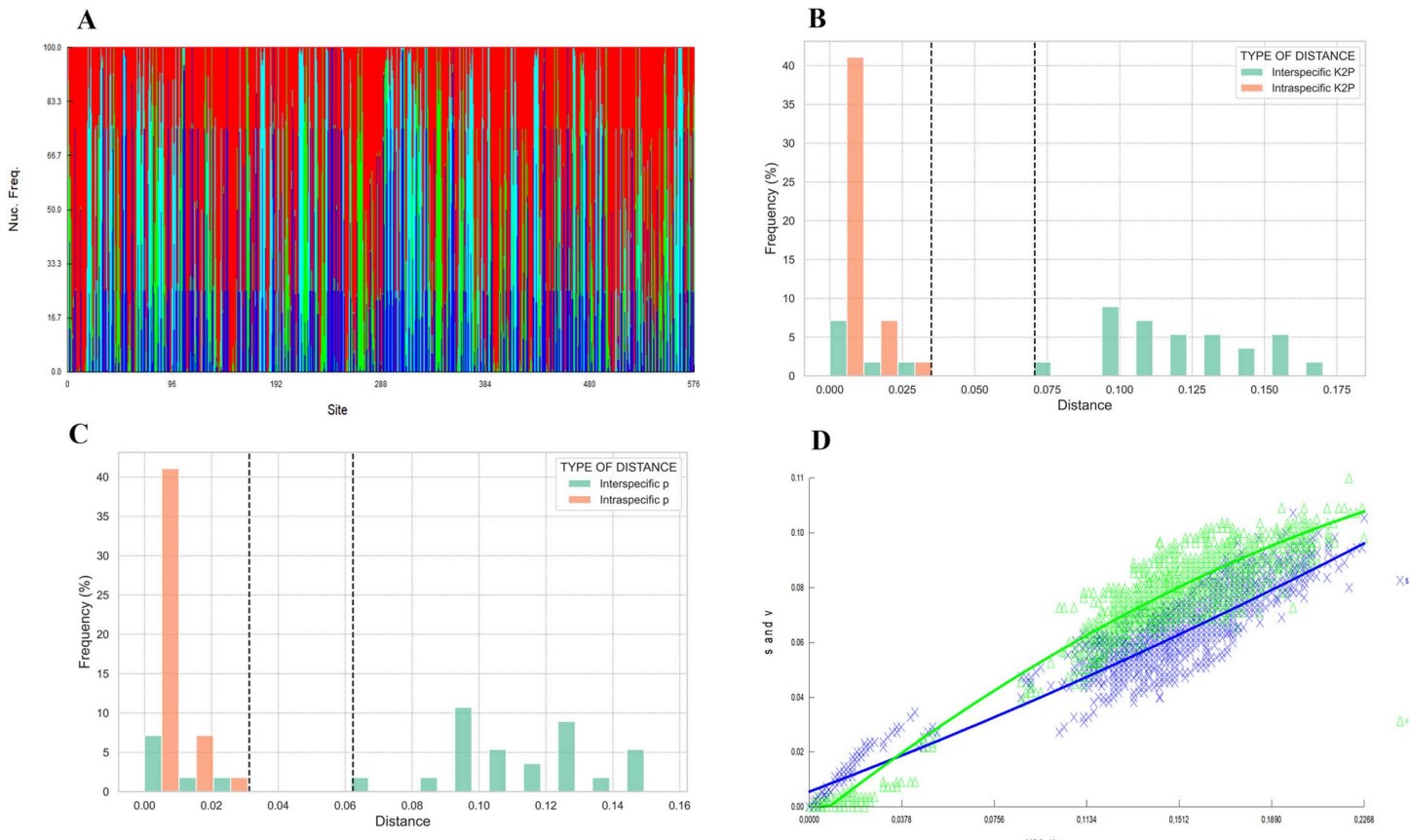

**Fig 4. Genetic divergence analysis from sand flies sequences collected in the departments of Amazonas and Caquetá.** (A) Nucleotide heterogeneity according to FINGERPRINT, (B) ABGD "barcode gap" K2P, (C) ABGD "barcode gap" uncorrected p distances. (D) Transitions/transversions rates versus genetic distance.

We detected 64 haplotypes (S2 Table), from 97 sequences, indicating a high degree of total nucleotide diversity per site ($\pi = 0.11596$; Standard deviation of $\pi = 0.00346$).

The number of haplotypes per species ranged from 1 to 9, the species with the greatest haplotype diversity (Hd) were *Ps. davisi* (N = 9; Hd: 0.9778), *Ps. ayrozai* (N = 5; Hd: 0.9333), *Th. cellulana* (N = 5; Hd: 0.593), *Ps. chagasi* (N = 4; Hd: 0.7143), and *Lu.* (*Hel.*) *tortura* (N = 4; Hd: 0.8000). In these species, the percentage of nucleotide diversity ranged from 0.1 to 2. 6% and 0.1 to 2.6% by the Pi and Pi (JC) models respectively (S2 Table). In contrast, *Ev.* (*Eva.*) *georgii*, *Ps. paraensis* and *Ty. witoto* presented unique haplotypes from different specimens (S2 Table).

The intraspecific *p* uncorrected and K2P genetic distances ranged from 0.0 to 2. 6% (average = 0.007; standard deviation = 0.002) and 0.0 to 2.6% (average = 0.007; standard deviation = 0.002), respectively (S2 Table). The species with the highest percentages of intraspecific distance were *Pa.* (*For.*) *aragaoi* (1. 2% p; 1. 2% K2P), *Ps. ayrozai* (1. 6% p; 1. 6% K2P), *Ps. davisi* (2.6% p; 2.6% K2P), *Th. howardi* (1.5% p; 1.5% K2P), and *Ev.* (*Ald.*) *walkeri* (1.3% p; 1.3% K2P). The minimum interspecific distance (NN) for each species ranged from 0.2% to 15.5% (p distances) and 0.2% to 16.6% (K2P distances) (S2 Table).

The genetic divergence analysis using the Automatic Barcode Gap Discovery ABGD method with the K2P model exhibited a marked "barcode gap" between the intra and interspecific distances of the sand flies collected in the Amazon, ranging from 0.25 to 0.75. This result suggests a clear separation between most of the species analyzed, as a delimiting criterion and indicator with an interspecific distance range from 0.075 to 0.175 (Fig 4B-4C). There is a slight overlap in the intraspecific distances (average 0.0042) associated with *Th. cellulana*, *Th. howardi* and *Th. velezbernali* which demonstrates limitations in the complete delimitation of these species and the need to use alternative markers with different replacement rate and inheritance. The transition/transversion rates versus genetic distance graph (Fig 4D) shows a grouping of transversions as genetic distance increases, and a pattern of substitutions that contributes to species differentiation, supporting interspecific separation.

### 3.4 Dendrogram constructed by the Neighbor-joining method (NJ), showing the phylogenetic relationship between species divided by genera

Intra-specific taxonomic association, that is clusters of sequences of the same species and redistribution and/or rearrangement of these clusters based on genetic divergence of different genera and/or other taxonomic levels of, in this case, Amazon sand flies, was obtained by constructing several dendrograms of NJ using 97 COI sequences from our study and 121 reference sequences available at GenBank. For a more detailed analysis, NJ was estimated by genera.

The NJ shows a correct specific clustering supported by high Bootstrap values, between 95 and 100%, for the genus *P*sychodopygus (Fig 5). The formation of two external clusters is evident. The first one formed by *Ps. davisi* that presented greater variability, along with the species *Ps. amazonensis*, *Ps. panamensis*, *Ps. wellcomei*, *Ps. chagasi*, and *Ps. paraensis*, while the second was formed by haplotypes of *Ps. ayrozai* (Fig 5).

In *Nyssomyia*, lower Bootstrap values were also found, between 85 and 96%, and a main external cluster was formed including *Ny. antunesi*, *Ny. yuilli pajoti* and *Ny. fraihai*, while *Ny. yuilli yuilli* is shown in a different cluster (Fig 6).

The NJ dendrogram that integrated *Evandromyia* and *Psathyromyia* shows Bootstrap values of 100% for the grouping of all species (Fig 7), which demonstrates a clear consistency with the morphological allocation and low genetic variability of these species. Internally, a cluster was formed for *Evandromyia* (Bootstrap 81%) and two groups for *Psathyromyia*, one for *Pa.* (*For.*) *aragaoi* and another for *Pa.* (*Psa.*) *dendrophyla*, both with 100% support.

The dendrogram clustering species with morphologically indistinguishable females, shows the formation of two external clusters, the first of *Trichophoromyia* and the second *Trichopygomyia* represented only by *Ty. witoto* (Fig 8). Internally, the first cluster is divided into two groups. The first comprising *Th. cellulana*, *Th. velezbernali* sequences with high genetic variability (Bootstrap of 88%) while *Th. howardi* is grouped in the second with Bootstrap values of 87% (Fig 8).

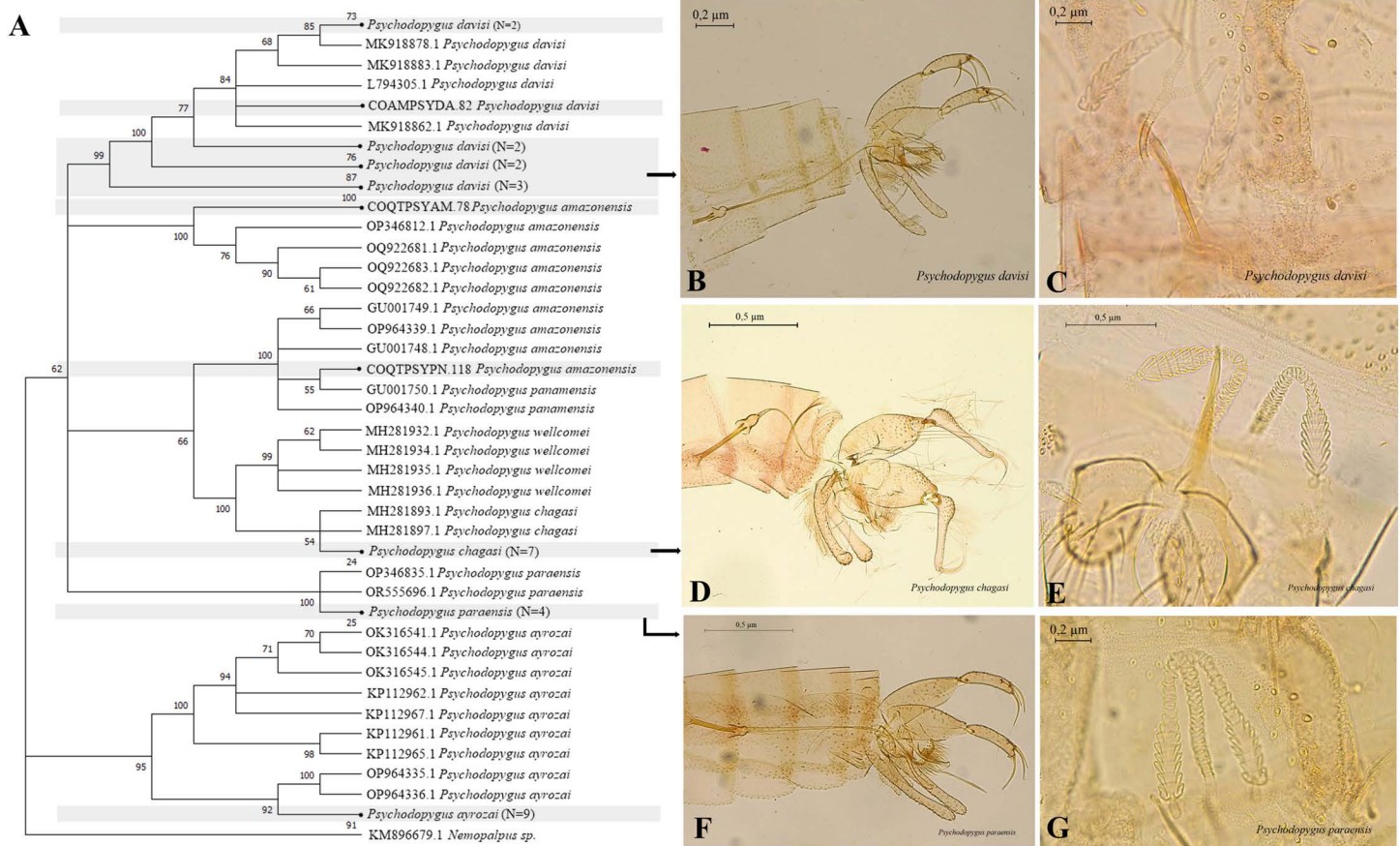

**Fig 5. Neighbor-joining gene tree and morphology of sand flies COI barcode sequences, with emphasis on *Psychodopygus*.** (A) NJ Dendrogram. (B-C) Male and female genitalia of Ps. davisi. (D-E) Male and female genitalia of Ps. chagasi. (F-G) Male and female genitalia of *Ps. paraensis*.

The dendrogram grouping the genera *Lutzomyia* and *Sciopemyia* (Fig 9) shows three outer clusters, all with Bootstrap support values of 100%. The first cluster comprises sequences of *Lu.* (*Tri.*) *sherlocki*, the second of *Lu.* (*Hel.*) *tortura*, and the third includes the *Sciopemyia* group, which is subdivided into three well-defined clusters: the first one corresponding to *Sc. fluviatilis*, the second to *Sc. preclara*, and the third to *Sc. sordellii*, each with Bootstrap supports of 100%.

The NJ dendrogram with genera of low abundances (Fig 10) clusters well-defined species with Bootstrap values of 100%, with evident low genetic variability, demonstrating clear consistency with morphological allocation. Two external clusters were formed, the first comprising sequences of *Pintomyia*, *Brumptomyia* and *Micropygomyia*, while the second was formed by *Vi. furcata* and *Vi. tuberculata*.

### 3.5 Identification of feeding sources

A total of 59 females out of the 755 collected presented visible traces of blood ingestion in the abdomen (7.8%). *Ev.* (*Ald.*) *walkeri* was the species with the highest number of females with evident blood intake (n = 17; 28.8%), followed by *Ny. yuilli pajoti* (n = 11; 18.6%), *Th. cellulana* (n = 8; 13.6%) and *Ny. fraihai* (n = 6; 10.2%). It was possible to detect the blood source in 12 of the 13 species analyzed, and the blood source was characterized in 39 females with the Cytb molecular marker (S3 Table) and 29 females (92.3%) with the marker rDNA 12S (S4 Table).

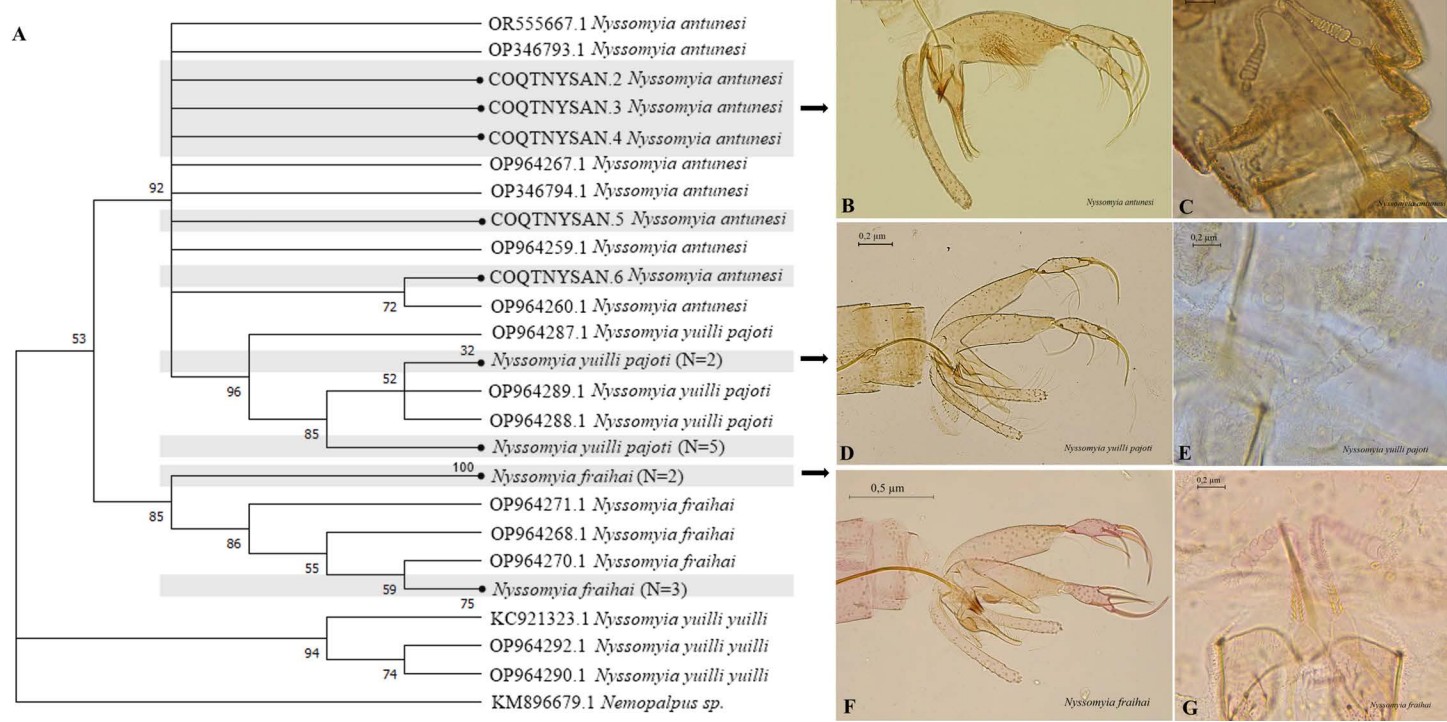

**Fig 6. Neighbor-joining gene tree and morphology of sand flies COI barcode sequences, with emphasis on *Nyssomyia*.** (A) NJ Dendrogram. (B-C) Male and female genitalia of *Ny. antunesi*. (D-E) Male and female genitalia of *Ny. yuilli pajoti*. (F-G) Male and female genitalia of *Ny. fraihai*.

After manual editing, the size of the fragment sequenced with the 12S rDNA marker ranged from 150 to 155 bps, while the length of the Cytb sequences ranged from 280 to 345 bps. The identity percentage by using the two markers was greater than 98% with a coverage of 91–100%, except for the *Bos taurus* (cow) and *Cheracebus lugens* (white-chested Titi), with 85.9 and 82.0% of identity, respectively (S3-S4 Tables).

The dendrogram generated from the Cytb marker reveals four clearly defined clusters (Bootstrap support of 100%), which highlight the genetic relationships and associated blood sources. The first cluster groups sequences (n = 17, species = 7) corresponding to *H. sapiens*, confirming its role as a frequent source of blood. The second cluster, with a is related to *Dasyprocta leporina* (brazilian aguti) (n = 1, species = 1) and *S. scrofa* (n = 10, species = 3). Finally, the third cluster, composed of *B. taurus* (n = 1, species = 1) (Fig 11).

The dendrogram grouping the vertebrates identified by the 12S marker reveals the formation of two main clusters, supported by Bootstrap values of 65% and 100%, respectively (Fig 12). The first cluster groups genetically close samples, such as *S. scrofa* and *H. sapiens* (n = 3, species = 2) with a Bootstrap value of 99%, evidencing genetic variability within each group. The second main cluster includes samples associated with *H. sapiens* (n = 11, species = 7) as well as *C. lugens* (n = 1, species = 1), *Saimiri macrodon* (ecuadorian squirrel monkey) (n = 1, species = 1) and *F. catus* (n = 1, species = 1).

*H. sapiens* was the most common blood source detected in 10 of the 12 species of sand flies analyzed, followed by *S. scrofa* detected in *Ev.* (*Ald.*) *walkeri*, *Ny. fraihai*, and *Th. cellulana*. DNA of *B. taurus* and *F. catus* was also detected in the latter two. In addition, mixed feeding (*H. sapiens*/*D. leporina*) was evidenced in a specimen of *Pi.* (*Pif.*) *nevesi* and *S. scrofa*/*C. lugens* in a specimen of *Ev.* (*Ald.*) *walkeri*, while *S. macrodon* was the only blood source detected in *Ps. davisi* (Fig 13).

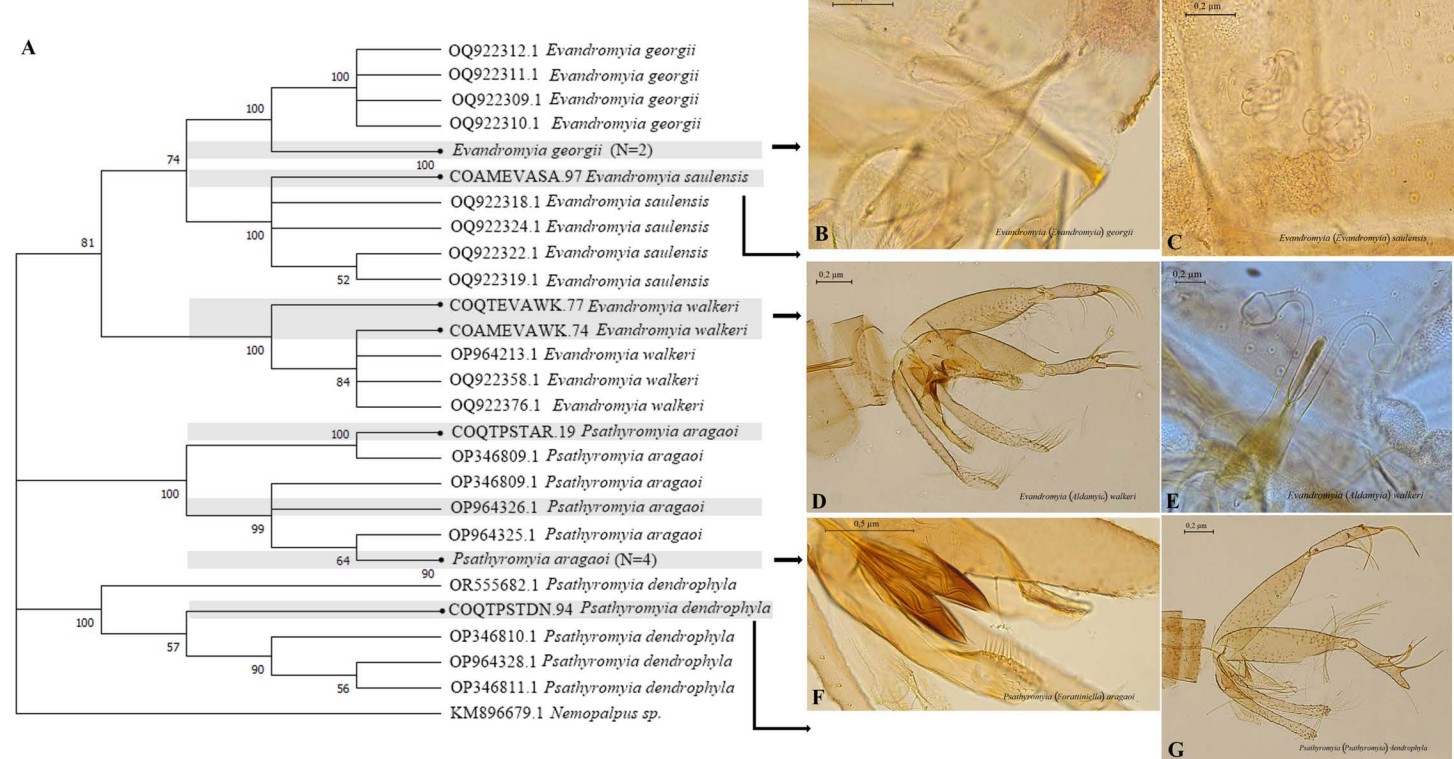

**Fig 7. Neighbor-joining gene tree and morphology of sand flies COI barcode sequences, with emphasis on *Evandromyia* and *Psathyromyia*.**
(A) NJ Dendrogram. (B) female genitalia of *Ev.* (*Eva.*) *georgii*. (C) female genitalia of *Ev.* (*Eva.*) *saulensis*. (D-E) male and female genitalia of *Ev.* (*Ald.*) *walkeri*. (F) male genitalia of *Pa.* (*For.*) *aragaoi*. (G) male genitalia of *Pa.* (*Psa.*) *dendrophyla*.

### 3.6 Detection of *Leishmania*

*Leishmania* DNA detection was performed in 37.9% (n = 310) (S2 Fig) of the total of females captured (n = 816), belonging to 28 species previously identified by integrative taxonomy. Using the HSP-70 N marker, *Leishmania* DNA was amplified in 28 samples from individual females with a minimum infection rate (MIR) of 9.0% (S5 Table).

The species with the highest *Leishmania* detection was *Ny. fraihai* (n = 8, 26%), collected in the peri and extra-domicile of all the localities sampled in Caquetá and in the extra-domiciliary of San Pedro de los Lagos (S5 Table). All *Ny. fraihai* specimens detected with DNA from *S. scrofa* and *B. taurus* collected in Caquetá were positive as well for *Leishmania*. Additionally, the DNA of the parasite was detected in a female sand flies of the same species classified as unfed and that was collected in the peridomicile areas of Sebastopol (Fig 13).

*Th. cellulana* was the second species with the highest detection rate (n = 4, 1.3%) (S5 Table). In these sand flies specimens collected in the extradomiciliar environment of Jericó, *Leishmania* DNA was detected in females with blood traces of *H. sapiens* as well as in those without evidence of blood feeding (Fig 13). *Ny. yuilli pajoti* presented the same rate as *Th. cellulana* (n = 4, 1.3%) (S5 Table). It was possible to detect *Leishmania* DNA in specimens collected from Santo Domingo that were also fed with *H. sapiens* (Fig 13). *Ev.* (*Ald.*) *walkeri* presented parasite detection rates lower than 0.9% in three specimens of this sand fly collected in the extradomicile of Jericó the presence of *Leishmania* was detected, one of them with blood meal associated with *S. scrofa* (S5 Table). Finally, *Leishmania* DNA was present in a single specimen of *Pi.* (*Pif.*) *nevesi* (n = 1, 0.3%) collected in the extra-domiciliary environment of Santo Domingo (S5 Table), which also exhibited a mixed feeding of *D. leporina*/*H. sapiens* (Fig 13).

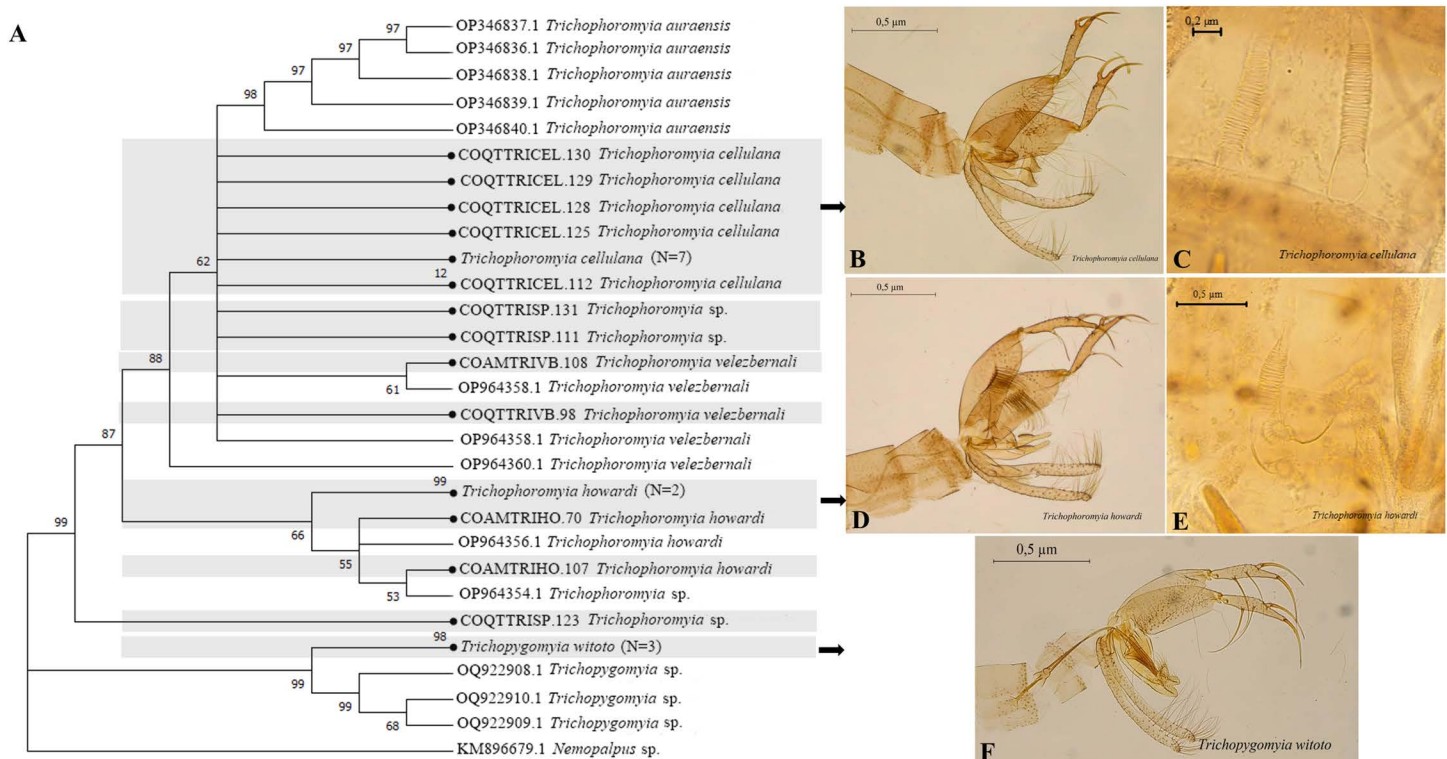

**Fig 8. Neighbor-joining gene tree and morphology of sand flies COI barcode sequences, with emphasis on *Trichophoromyia* and *Trichopy-gomyia*.** (A) NJ Dendrogram. (B-C) male and female genitalia of *Th. cellulana*. (D-E) male and female genitalia of *Th. howardi*. (F) male genitalia of *Ty. witoto*.

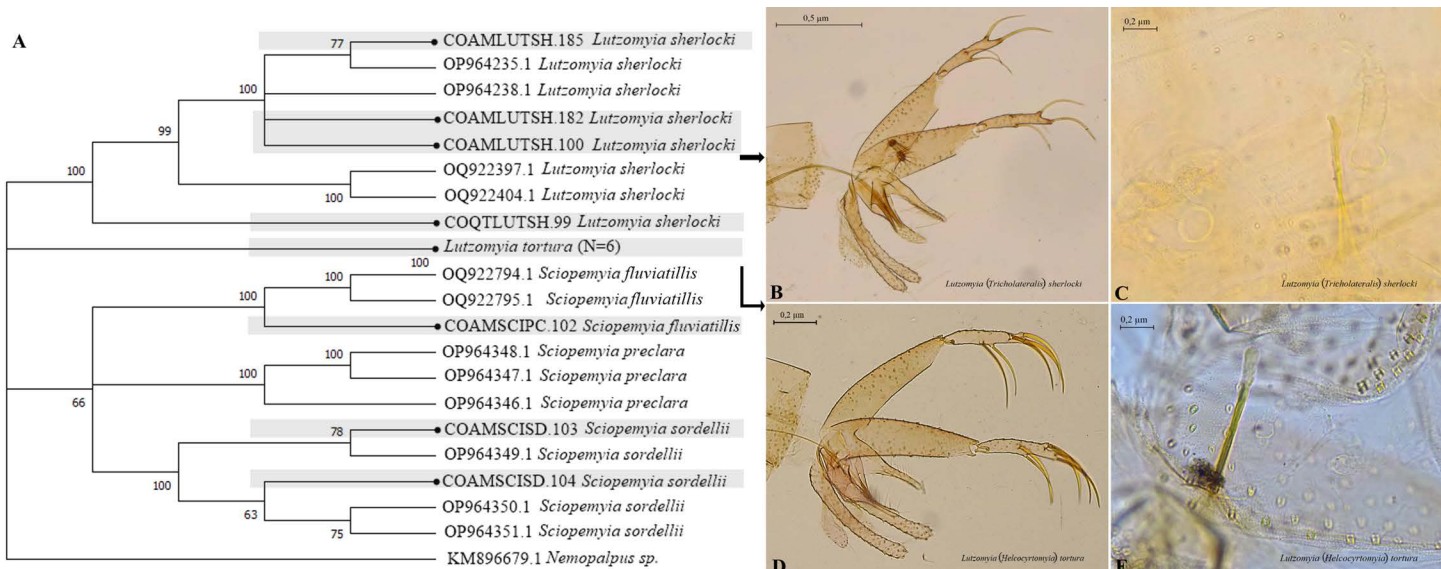

**Fig 9. Neighbor-joining gene tree and morphology of sand flies COI barcode sequences, with emphasis on *Lutzomyia* and *Sciopemyia*.** (A) NJ Dendrogram. (B-C) male and female genitalia of *Lu.* (*Trl.*) *sherlocki*. (D-E) male and female genitalia of *Lu.* (*Hel.*) *tortura*.

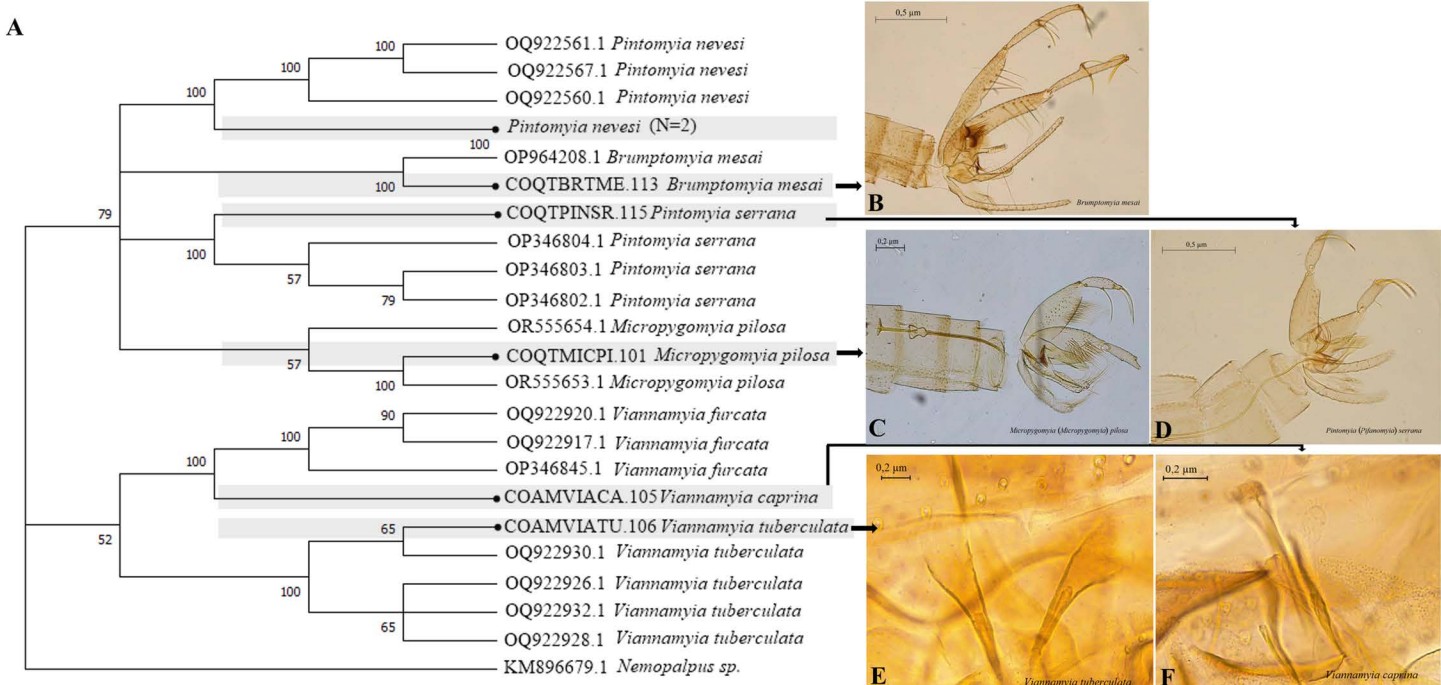

**Fig 10. Neighbor-joining gene tree and morphology of sand flies COI barcode sequences, with emphasis on genera of lesser abundance.** (B) male genitalia of *Br. mesai*. (C) male genitalia of *Mi.* (*Mic.*) *pilosa*. (D) male genitalia of *Pi.* (*Pif.*) *serrana*. (E) female genitalia of *Vi. tuberculata*. (F) female genitalia of *Vi. caprina*.

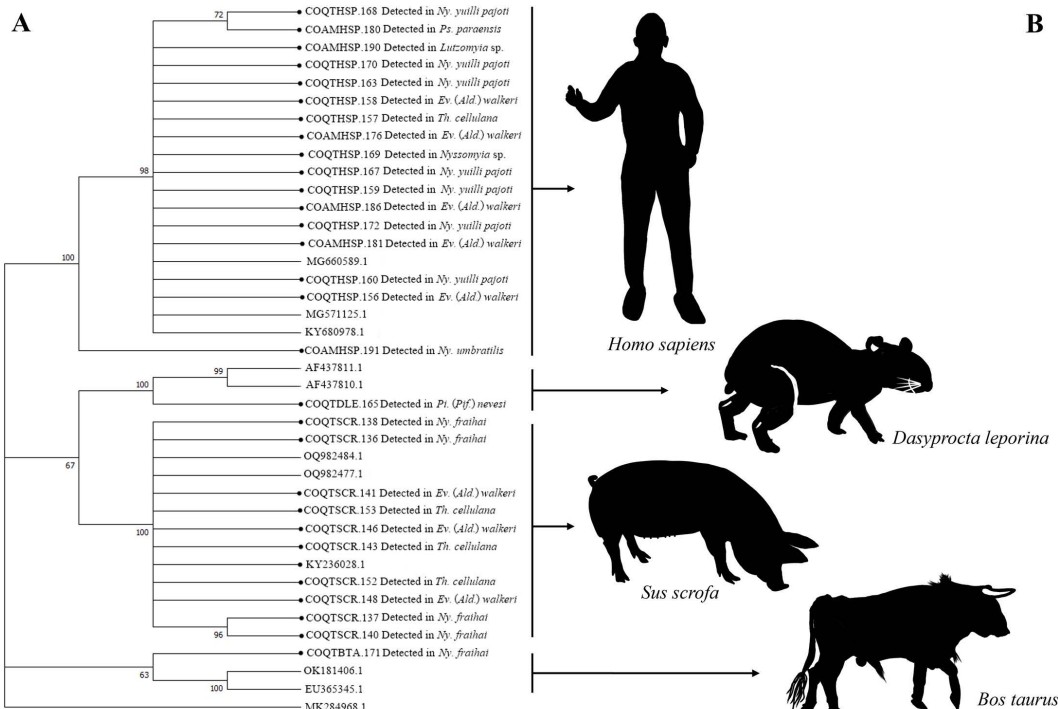

**Fig 11. Neighbor-joining gene tree and blood food sources detected in sand flies collected in the Colombian Amazon region, using the molecular marker Cytochrome B (Cytb).** (A) Dendrogram NJ. (B) Vertebrate species identified.

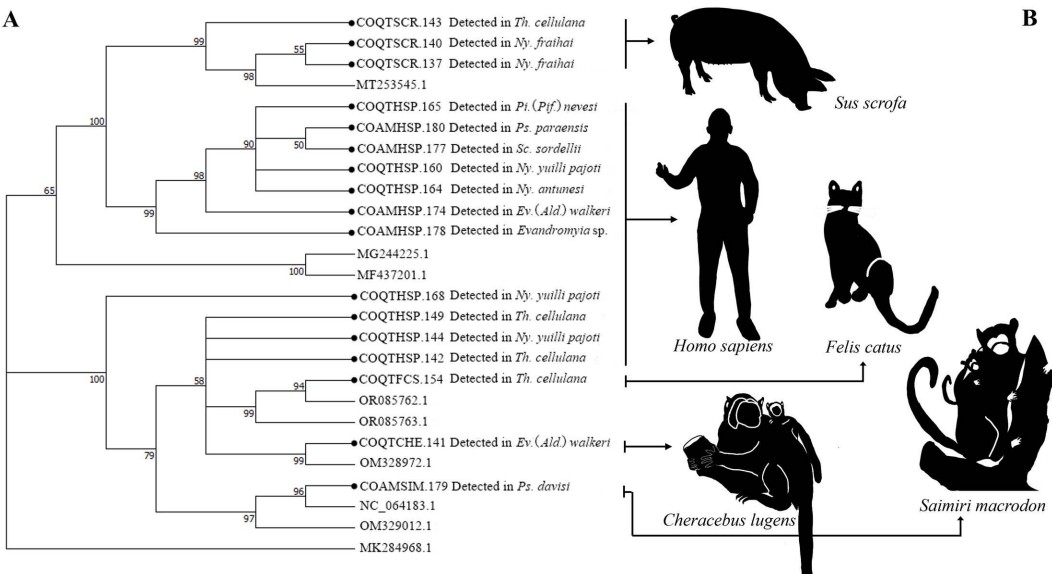

**Fig 12. Neighbor-joining gene tree and blood food sources detected in sand flies collected in the Colombian Amazon region, using the molecular marker 12S.** (A) Dendrogram NJ. (B) Vertebrate species identified.

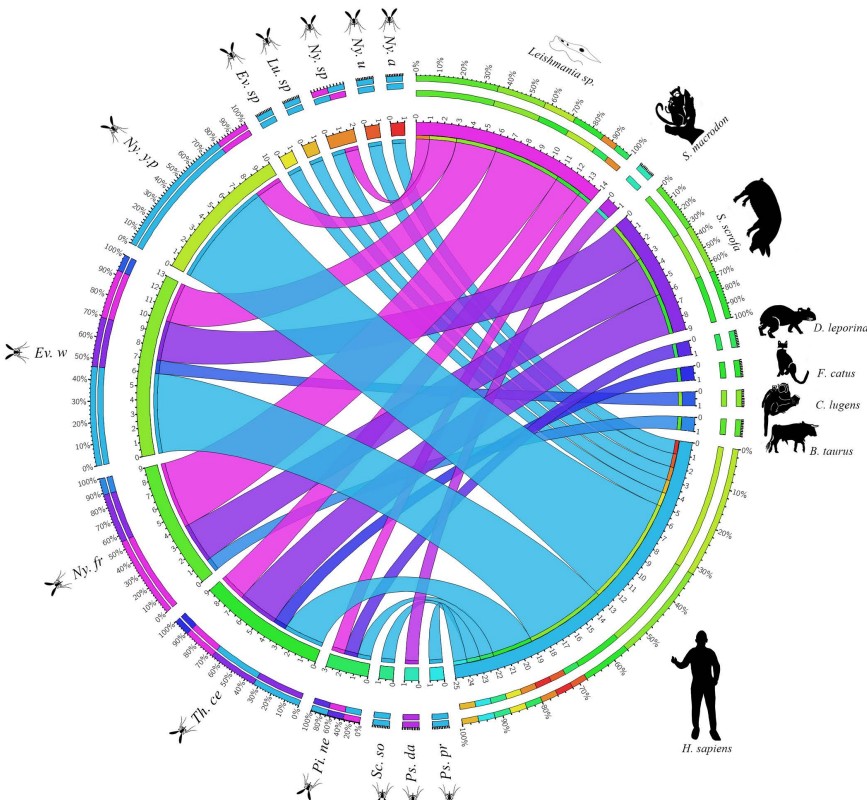

**Fig 13. Relationship between sources of blood ingestion and the detection of *Leishmania* DNA in sand flies from the department of Amazonas and Caquetá. *Ny.a*:** *Ny. antunesi,* ***Ny.u*:** *Ny. umbratilis,* ***Ny. yp*:** *Ny. yuilli pajoti,* ***Ev. w*:** *Ev. (Ald.) walkeri,* ***Ny. fr*:** *Ny. fraihai,* ***Th. ce*:** *Th. cellulana,* ***Pi. ne*:** *Pi. (Pif.) nevesi,* ***Sc. so:*** *Sc. sordellii,* ***Ps. da*:** *Ps. davisi,* ***Ps.pr*:** *Ps. paraensis.*

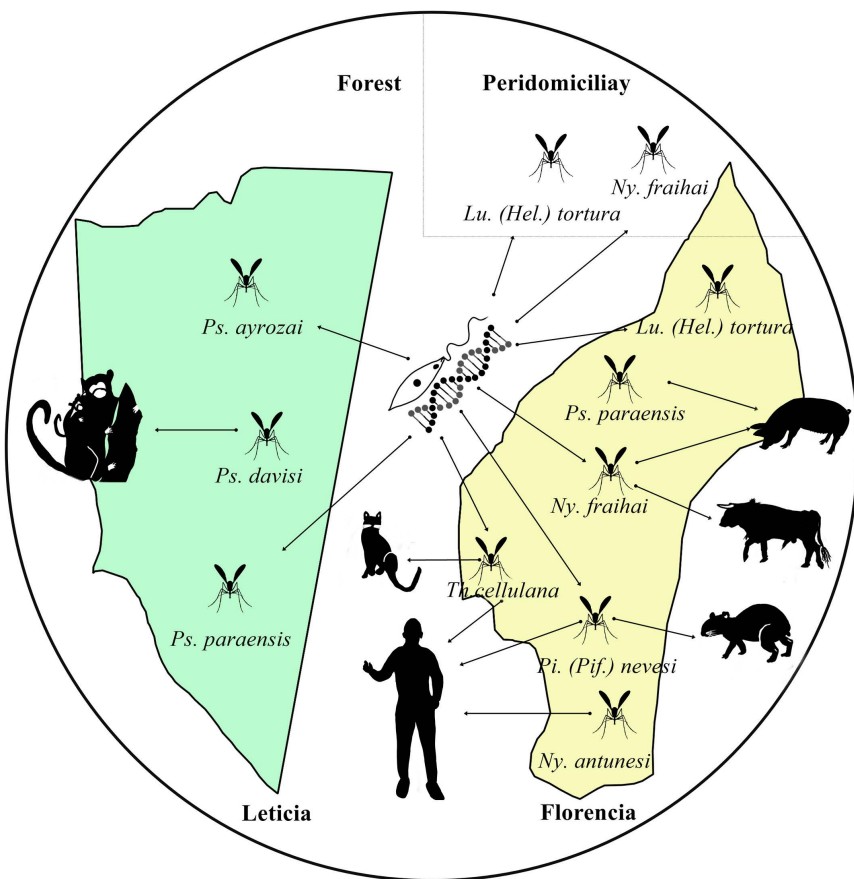

**Fig 14. The ecological dynamics of the tripartite relationship among sand flies, blood meal sources and *Leishmania* in forest and peridomiciliary areas in the Amazon region from Colombia.**

## 4. Discussion

This study integrates morphological, molecular, and ecological approaches for the identification of sand flies in the Colombian Amazon, a region of high biodiversity and great vulnerability to the effects of climate change and human disturbances [60]. To the best of our knowledge, this work presents the first DNA barcoding (COI) sequences for sand flies from the department of Caquetá and the second study of this type in Amazonas [29]. Sixty-four new haplotypes were detected and sequences of *Lu.* (*Hel.*) *tortura*, *Th. cellulana, Ty. witoto*, and *Vi. caprina* not previously analyzed with this marker are provided. In addition, three new circulation records are presented for Amazonas, *Vi. caprina*, *Vi. tuberculata* and *Sc. fluviatilis*, nine Caquetá, *Br. mesai*, *Ev.* (*Ald.*) *georgii*, *Ny. fraihai*, *Ny. yuilli pajoti*, *Lu.* (*Trl.*) *sherlocki*, *Lu.* (*Hel.*) *tortura*, *Ps. carrerai thula*, *Th. howardi* and *Pi.* (*Pif.*) *serrana*, and one for Colombia, *Sc. fluviatilis*, as far as we know. Ten species of epidemiological relevance were identified, seven vertebrate species that act as a source of blood ingestion and the presence of *Leishmania* sp. was detected in areas where cases of tegumentary leishmaniasis have historically been recorded.

### 4.1 Sand flies and epidemiological relevance

In this study, *Ev.* (*Ald.*) *walkeri* and *Ps. amazonensis* were the most abundant species in the Amazonas department, representing 36.6% of the total number of specimens. Despite limited information on these species in the region, previous studies that use a similar collection approach evidenced that *Ev.* (*Ald.*) *walkeri* does not present high dominance [30]. This can be attributed to the ecotope selected for collection, the number of traps used, and the sampling effort time.

The department of Caquetá showed a greater richness of species, with 22 species found in the different localities, being a 18.2% higher when compared to the amount of species collected in the Amazonas department, which can be attributed to a greater sampling effort and a wider geographical range of collection in the first case. The results of species richness in this department are congruent with those recorded at Araracuara, Caquetá, during 1981 where Morales and Minter collected 35 species of sand flies, including six new records for the country and one new for science [27].

In Caquetá, *Th*. *cellulana*, *Ny*. *yuilli pajoti* and *Ny*. *antunesi* are the most abundant species, representing 61.4% of the total number of specimens caught. These results differ from previous studies in the same region, where collection during rainy seasons showed low abundance of these species [27,61,62].

Since the description of *Th*. *cellulana* in Caquetá in 1979 [62], it had not been recorded in this department until the present study. To date, its distribution is exclusively Amazonian, including regions of the Colombian Amazon (departments of Guaviare and Putumayo) [63,64], Peruvian and Ecuadorian [65,66], where the reported abundance does not exceed 3% of the total composition. Although the epidemiological importance of this species has not yet been explored, the growing interest in the ecoepidemiological aspects associated with the genus *Trichophoromyia* has led to consider several of its species as suspected vectors [67]. Such is the case of two of the five species recorded in Caquetá, *Th*. *ubiquitalis*, found with flagellates of *L*. (*V*.) *lainsoni* in Brazil [68], and detected with DNA from *L*. (*V*.) *lainsoni*, *L*. (*L*.) *amazonensis* and *L*. (*V*.) sp. in Brazil and Ecuador [69–71]. And *Th*. *auraensis* has been found infected with DNA from *L*. (*V*.) *braziliensis*, *L*. (*V*.) *lainsoni* and *L*. (*V*.) *guyanensis* in Peru and Brazil [72,73].

*Ny*. *antunesi* is another abundant species in Caquetá that exhibits highly anthropophilic behavior [74,75] and has been found during leishmaniasis outbreaks in the Meta department (Colombian eastern plains) with DNA of *Leishmania* not genotyped [76]. In Brazil, this species has been found infected with *L*. (*V*.) *guyanensis* [75], and is recognized as a vector of *L*. (*L*.) *infantum* and *L*. (*V*.) *braziliensis* [77]. Other sand fly like *Ny*. *yuilli pajoti*, are recognized to have the widest geographical distribution in Colombia with presence in Guaviare, Putumayo, Guainía, and Amazonas departments [20], has been registered as an anthropophilic species that is highly abundant in areas of intra, peri and extra-domiciliary in Putumayo [64], and has been collected frequently in rainy seasons and with low abundance in Amazonas and Guaviare department of Colombia [30,63].

During this study, a new geographical distribution of various sand flies species was found. Previous records of *Vi*. *caprina* have been associated with its distribution in the Colombian Amazon, specifically in the Guaviare department [63] but not in the Amazonas department as found in this study. This sand fly is not classified as a vector considering that it is not involved in the transmission cycle while its distribution in the country also includes the Andean and Pacific regions [20]; *Vi*. *tuberculata* previously documented in Colombia (Caquetá and Guaviare), as well as in the Andean and Pacific regions of Colombia, in addition to the Orinoco region [20], and in the Amazon region of Brazil where it was found to be naturally infected with *L*. (*V*.) *utingensis* [78], and *Sc*. *fluviatilis*, whose presence was unknown in the country until now, presents a jungle-like distribution behavior, with distributions recorded in French Guiana and several states of the Brazilian Amazon [45].

Although flagellate DNA has been detected in *Sc*. *fluviatilis*, it has not yet been considered as a probable vector of *Leishmania* [79]. Within *Sciopemyia*, blood traces of *H*. *sapiens* were also detected in specimens of *Sc*. *sordellii*. The anthropophilic habit had been previously documented [80], as well as the preference for other vertebrates, including *C*. *lupus*, *G*. *gallus*, *S*. *scrofa*, *Bokermannohyla martinsi* and *Rattus rattus* (black rat) [81–84] and the presence of DNA from *L*. (*V*.) *guyanensis*, *L*. (*L*.) *infantum* and *L*. (*V*.) *braziliensis* in specimens from different areas of Brazil [80,85,86]. These findings highlight the need to investigate the possible role of *Sc*. *sordellii* in the transmission of *Leishmania* in Amazonia, as well as to deepen the dimension of its feeding plasticity, key aspects for its survival and ecology.

Among the nine species reported for the first time in the department of Caquetá, two represent, to our knowledge, the first record for the Colombian Amazon region, *Pi*. (*Pif*.) *serrana*, an anthropophilic species previously recorded in the Andean, Caribbean, and Orinoco regions [20], is associated with an endemic zone of *L*. (*V*.) *braziliensis* in Norte de

Santander [87], while other species like *Ps. carrerai thula*, previously recorded in the Pacific and Andean regions of the country, is not currently associated with *Leishmania* transmission [20,88].

In addition to the species previously recorded with epidemiological relevance due to their proven or suspected role in the transmission of *Leishmania* parasites, the presence of *Ny. umbratilis*, the main vector of *L. (V.) guyanensis* in Brazil [89], is noteworthy. In Colombia, no studies have examined its role as a vector, however, it is closely associated with *L. (V.) guyanensis* in the Amazonas [90]. Currently, there are no records in Colombia of *Leishmania* infection in *Ny. fraihai*, however, this species has been detected with trypanosomatids of the genus *Endotrypanum* in Porto Velho, Brazil [91]. In the case of *Lu. (Tri.) sherlocki*, captured in Caquetá and Amazonas in this study, there are records of this species in an endemic area for CL and visceral leishmaniasis (VL) in Pará, Brazil [92] and detected with *Leishmania* sp. DNA in a highly endemic area for leishmaniasis in the Peruvian Amazon [93]. Sand flies like *Lu. (Hel.) tortura*, have been reported in eight municipalities in the department of Putumayo and in Puerto Nariño locality, in the Amazonas department [30,64,94]. This species is highly anthropophilic, and similar to this study, it has been captured in the peri and extradomiciliary areas [64,95]. It has not been detected with *Leishmania* DNA in the Colombian territory, but in 2013 Kato and collaborators detected high rates of natural infection by *L. (V.) naiffi* suggesting its participation in the transmission cycle of this parasite in the Ecuadorian Amazon region [95].

The significance of finding sand flies like *Ps. ayrozai*, *Ps. davisi*, and *Ps. amazonensis* relies on the fact that they have been detected with *L. (V.) naiffi* in some regions of Brazil [96]. Specifically, *Ps. ayrozai* has also been detected with DNA from *L. (L.) amazonensis*, *L. (V.) braziliensis*, and *L. (V.) guyanensis* [80,97]. *Ps. panamensis*, associated with cutaneous leishmaniasis outbreaks, was recently classified as a level 4 vector [98], is highly anthropophilic, predominates in intra and peridomiciliary areas [99], and has been detected with DNA from *L. (V.) panamensis* [99] and *L. (L.) amazonensis* [100]. *Ps. chagasi* is of epidemiological relevance in Brazil and recently, DNA from *L. (V.) braziliensis* and *L. (V.) lainsoni* was detected in the municipality of Itapuã [97]. Although the role of *Ev. (Ald.) walkeri* in the transmission of leishmaniasis in Colombia has not been conclusively demonstrated, DNA from *L. (V.) braziliensis* [13] and *L. (V.) guyanensis* [75] has been detected in Acre state, Brazil.

## 4.2 Barcoding, a tool for species identification sand flies

Genetic analysis of the COI gene made it possible to differentiate species where the minimal macrometric differences of the males, associated with their parameres, require considerable expertise for their identification, as is the case with *Ny. yuilli pajoti* and *Ny. fraihai*. Improvements in the Barcode library allowed the consolidation of taxonomy within the genera *Trichophoromyia* and *Trichopygomyia*, facilitating the differentiation of species with isomorphic females, mainly *Th. cellulana*, *Th. howardi*, and *Ty. witoto*. The use of COI to differentiate species is justified by clear intraspecific (2.6%) and interspecific (16.6%) genetic distances. In some species with a considerable degree of morphological differentiation, as in the case of genera like *Nyssomyia*, *Evandromyia* and *Psychodopygus* may indicate the presence of a "barcode gap" within species evidenced by the clear separation into two groups in the NJ.

However, when species are closely related, the analysis of genetic distances may imply an overlapping of data, preventing the recognition of a clear gap, as occurred in this study with *Th. cellulana* and *Th. velezbernali*. This is consistent with the evidence reported for species of the genus *Trichophoromyia* collected in the department of Amazonas [29]. The limitation of barcoding to discriminate sand flies species has been documented in several cases, especially when genetic distances overlap due to a possible relatively recent divergence as reported in *Ev. (Ald.) carmelinoi*, *Ev. (Ald.) evandroi* and *Ev. (Ald.) lenti* [101] or *Ps. complexus*, and *Ps. wellcomei* [102]. This type of integrative taxonomy studies can be complemented with the use of multilocus tools that include nuclear markers such as the Second Inward Transcribed Spacer (ITS2) or Elongation Factor alpha 1 (Ef1-α) [103], in addition to other markers such as cytochrome b (Cytb) and the 28S ribosomal RNA gene [104] that can be essential for the complete delimitation of taxa, especially for those closely related [105,106].

### 4.3 Hosts and *Leishmania* in the Amazon region

Despite the challenges posed by the disturbance of the Colombian Amazon rainforest, such as population growth, deforestation, and environmental changes due to climate change [60], rural areas still favor the conservation of biodiversity and the richness of phlebotomines, whose wild behavior continues to shape their distribution and feeding habits. Among the vertebrate species that are considered as sand flies blood meal sources found in the Colombian Amazon localities analyzed, *H. sapiens* was identified as a blood source of *Ny. antunesi*, which is congruent with previous records documenting its anthropophilic behavior [16], while there are also reports of its affinity for other mammals such as *B. taurus*, *Choloepus didactylus* (sloth), *D. leporina*, *S. scrofa*, and *T. tetradactyla* and birds such as *Pteroglossus aracari* (toucan) [16]. *H. sapiens* was also the only blood meal source associated with *Ps. panamensis* in this study, however other investigations have found a variety of blood traces in this sand fly, including *Marmosa robinsoni* (robinson's mouse) [107] and *G. gallus* [19,107]. In addition, this species has been reported with mixed feedings of *H. sapiens*/*Equus asinus*, *H. sapiens*/*B. taurus*, *B. taurus*/*C. familiaris*, and *S. scrofa*/*H. sapiens* [108,109].

The anthropophilic behavior of *Ev.* (*Ald.*) *walkeri* collected in the two departments studied was evidenced by the detection of blood traces of *H. sapiens*. Traces of *C. lugens* and *S. scrofa* were also identified in this species; previously in Brazil it was found with blood traces of *G. gallus* [13] and *S. scrofa* [110], while the presence of *Leishmania* sp. in the specimens collected from Caquetá is not strange, as there are records of natural *Leishmania* sp. infection in Peru [93] and *L.* (*V.*) *braziliensis* infection in Brazil [13]. Although information on the ecology of *Th. cellulana* is scarce, this study identified the presence of DNA from *H. sapiens*, *S. scrofa* and *F. catus*, suggesting anthropophilic and eclectic behavior. In particular, the presence of *F. catus* DNA is relevant, as these felids have been reported naturally infected and categorized as active carriers of *L.* (*L.*) *infantum* in several regions of Brazil, where the transmissibility of the parasite to the vector *Lu.* (*Lut.*) *longipalpis* has categorized them as key players in the VL cycle [111,112]. In addition, *Leishmania* sp. DNA was detected, reinforcing the connection between sources of ingestion and possible infection. Although not previously recorded in this species, this finding underscores the importance of focusing future research on ecoepidemiological aspects that will provide accurate information on the possible role of *Th. cellulana* in *Leishmania* transmission.

The presence of *Leishmania* in *Pi.* (*Pif.*) *nevesi* is consistent with previous studies that identified *L.* (*V.*) *braziliensis* as the pathogen involved in the infection [13]. Although blood traces of *H. sapiens* and *G. gallus* have been detected previously in this species [13,80], this study highlights the ability of this species to feed even on different blood sources. Blood traces found in *Ny. umbratilis*, a recognized vector of *L.* (*V.*) *guyanensis* in several countries including Brazil [113], corresponded with *H. sapiens*, demonstrating its anthropophilic behavior. Although no *Leishmania* DNA was detected in this study, previously, parasites of *L.* (*V.*) *guyanensis* were isolated from females collected in the department of Amazonas [98], In addition, *Ny. umbratilis*, has been detected with DNA from *Endotrypanum* sp. [80] and with blood traces of *Plecturocebus bernhardi* and *S. scrofa* mainly in the Brazilian Amazon [110].

*Ps. davisi* previously detected with DNA from *L.* (*V.*) *braziliensis* [13] and *Endotrypanum* sp. [80], was not found positive with parasite DNA in this study, although its blood source preferences include *Pecari tajacu*, *S. scrofa* and *T. tetradactyla* as evidenced in other studies [110], in this research only blood traces from *H. sapiens* were found. The finding of DNA of *Leishmania* not genotyped in understudied species such as *Ny. yuilli pajoti* and *Ny. fraihai* provides valuable information on their ecology, highlighting their anthropophilic habits and their interaction with humans in peri and extra-domiciliary areas. The selective preference of some sand flies species for certain hosts, such as *S. scrofa* in the Jericó locality, raises questions about their role in the dynamics of the pathogen, especially when DNA of the parasite was detected in sand flies associated with this vertebrate.

Although in this study it was not possible to perform consistent sequencing of the *Leishmania* amplicons, possibly due to DNA degradation or low concentration. Future studies contemplate the use of more sensitive techniques, such as ITS1–2 markers, nested PCR, qPCR and NGS, for accurate characterization of circulating species.

## 5. Conclusions

The results presented provide a solid information for future research on how the tripartite interactions between sand flies, hosts, and pathogens including the emergence and increased abundance of opportunistic hematophagous species (S3 Fig). Furthermore, these findings underscore the urgent need to protect remaining habitats and mitigate anthropogenic impacts that threaten to significantly alter the ecological dynamics of these areas.

In conclusion, the tripartite relationship among insects, hosts, and *Leishmania* is crucial for understanding transmission dynamics, especially in poorly studied areas of the Amazon with a high prevalence of leishmaniasis. These findings provide valuable information on zoonotic cycles in the region. The identification of new blood sources suggests that some animals may be underestimated as potential reservoirs of *Leishmania* (Fig 14). This study reinforces the relevance of the "One Health" approach, which integrates human, animal, and environmental health, emphasizing the need to monitor sand flies while preserving biodiversity.

## Supporting information

**S1 Fig. Alpha Diversity Indices Across Localities in the Colombian Amazon (Amazonas and Caquetá).**
(TIF)

**S2 Fig. Partial amplification of the HSP-70N gene Molecular for detections of *Leishmania* sp. in DNA total of sand flies collected in the Colombian Amazon region.**
(TIF)

**S3 Fig. Graphical summary of the main results.**
(TIF)

**S1 Table. Ecological estimates of abundance (standardized index), constancy, and dominance of sand flies species from Amazonas and Caquetá.**
(XLSX)

**S2 Table. Sand flies species, number of individuals with mitochondrial haplotypes, maximum intraspecific genetic divergence and minimum distance to nearest neighbor of the Colombian Amazonian sand flies species analyzed in this study.**
(DOCX)

**S3 Table. Vertebrates acting as sources of blood ingestion in sand flies detected with the molecular marker Ctyb in species collected in Amazonas and Caquetá, Colombian Amazon region.**
(DOCX)

**S4 Table. Vertebrates acting as sources of blood ingestion in sand flies detected with the molecular marker 12S in species collected in Amazonas and Caquetá, Colombian Amazon region.**
(DOCX)

**S5 Table. Prevalence of *Leishmania* infection and minimum infection rate (MIR) in sand flies collected in the Colombian Amazon region.**
(DOCX)

## Acknowledgments

We gratefully acknowledge the communities of the different localities, to Lina Marcela Manjarrez and her team of assistants trained in community work of the Health of Florencia, Caquetá, to Dr. Luz Mila Murcia Montaño, from the Amazonas Government and the Amazonas Public Health Study Group (GESPA), to Biologist and Master's student Jennifer Danitza

Viafara and Biological Engineer, to Master's student Alejandro Castañeda M, for their invaluable support during sand flies collections, and the access to their laboratories. We express our special gratitude to the Program for the Programa de Estudio y Control de Enfermedades Tropicales (PECET), to its director Dr. Sara M. Robledo and Dr. Laura Cristina Posada-López for providing the space for the identification of sand flies, as well as for the donation of insects from the *Ae. aegypti* colony Rockefeller and *Leishmania* strains used as controls in this study.

## Author contributions

**Conceptualization:** Katerine Caviedes-Triana, Daniela Duque-Granda, Gloria Cadavid-Restrepo, Claudia X. Moreno-Herrera, Rafael Vivero-Gomez.

**Data curation:** Katerine Caviedes-Triana, Rafael Vivero-Gomez.

**Formal analysis:** Katerine Caviedes-Triana, Rafael Vivero-Gomez.

**Funding acquisition:** Gloria Cadavid-Restrepo, Claudia X. Moreno-Herrera.

**Investigation:** Katerine Caviedes-Triana, Daniela Duque-Granda, Claudia X. Moreno-Herrera, Rafael Vivero-Gomez.

**Methodology:** Katerine Caviedes-Triana, Daniela Duque-Granda, Claudia X. Moreno-Herrera, Rafael Vivero-Gomez.

**Project administration:** Gloria Cadavid-Restrepo, Claudia X. Moreno-Herrera.

**Resources:** Gloria Cadavid-Restrepo, Claudia X. Moreno-Herrera, Rafael Vivero-Gomez.

**Software:** Katerine Caviedes-Triana, Rafael Vivero-Gomez.

**Supervision:** Gloria Cadavid-Restrepo, Claudia X. Moreno-Herrera, Rafael Vivero-Gomez.

**Validation:** Katerine Caviedes-Triana, Gloria Cadavid-Restrepo, Claudia X. Moreno-Herrera, Rafael Vivero-Gomez.

**Visualization:** Katerine Caviedes-Triana, Daniela Duque-Granda, Claudia X. Moreno-Herrera, Rafael Vivero-Gomez.

**Writing – original draft:** Katerine Caviedes-Triana, Rafael Vivero-Gomez.

**Writing – review & editing:** Katerine Caviedes-Triana, Daniela Duque-Granda, Gloria Cadavid-Restrepo, Claudia X. Moreno-Herrera, Rafael Vivero-Gomez.

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
