## [Decision Letter · Decision Letter 0]

25 May 2025

Dear Dr. Moreno-Herrera,

Response to Reviewers
Revised Manuscript with Track Changes
Manuscript

Shaden Kamhawi

co-Editor-in-Chief

Paul Brindley

co-Editor-in-Chief

**Additional Editor Comments :**

**Journal Requirements:**

At this stage, the following Authors/Authors require contributions: Katerine Caviedes-Triana, Daniela Duque-Granda, Gloria Cadavid-Restrepo, Claudia Ximena X. Moreno-Herrera, and Rafael Vivero-Gomez. Please ensure that the full contributions of each author are acknowledged in the "Add/Edit/Remove Authors" section of our submission form.

- ® on pages: 1, and 6.

4) Please ensure that all Figure files have corresponding citations and legends within the manuscript. Currently, Figure 15 in your submission file inventory does not have an in-text citation. If the figure is no longer to be included as part of the submission, please remove it from the file inventory.

 Note: The Figure legends should not be uploaded as separate files in the online submission form, they should be included in the manuscript.

5) Tables should not be uploaded as individual files. Please remove Table 1 file and just include it in your manuscript file as an editable, cell-based object. For more information about how to format tables, see our guidelines:

https://journals.plos.org/plosntds/s/tables 

Potential Copyright Issues:

i) Please confirm (a) that you are the photographer of Graphical Abstract, 5, 6, 7, 8, 9, and 10, or (b) provide written permission from the photographer to publish the photo(s) under our CC BY 4.0 license.

ii) Figures Graphical Abstract, 11B, 12B, 13, and 15. Please confirm whether you drew the images / clip-art within the figure panels by hand. If you did not draw the images, please provide (a) a link to the source of the images or icons and their license / terms of use; or (b) written permission from the copyright holder to publish the images or icons under our CC BY 4.0 license. Alternatively, you may replace the images with open source alternatives. See these open source resources you may use to replace images / clip-art:

iii) Figure 1. Please (a) provide a direct link to the base layer of the map (i.e., the country or region border shape) and ensure this is also included in the figure legend; and (b) provide a link to the terms of use / license information for the base layer image or shapefile. We cannot publish proprietary or copyrighted maps (e.g. Google Maps, Mapquest) and the terms of use for your map base layer must be compatible with our CC BY 4.0 license.

7) Please note that your Data Availability Statement in the online submission form is currently missing the DOI/accession number of each dataset OR a direct link to access each dataset. If your manuscript is accepted for publication, you will be asked to provide these details on a very short timeline. We therefore suggest that you provide this information now, though we will not hold up the peer review process if you are unable.

8) Please amend your detailed Financial Disclosure statement. This is published with the article. It must therefore be completed in full sentences and contain the exact wording you wish to be published.

3) If any authors received a salary from any of your funders, please state which authors and which funders.

9) Please provide a completed 'Competing Interests' statement, including any COIs declared by your co-authors. If you have no competing interests to declare, please state "The authors have declared that no competing interests exist".

**Comments to the Authors:**

**Please note that three reviews are uploaded as attachments.**

**Reviewers' comments:**

**Key Review Criteria Required for Acceptance?**

**Methods**

-Are the objectives of the study clearly articulated with a clear testable hypothesis stated?

-Is the study design appropriate to address the stated objectives?

-Is the population clearly described and appropriate for the hypothesis being tested?

-Is the sample size sufficient to ensure adequate power to address the hypothesis being tested?

-Were correct statistical analysis used to support conclusions?

-Are there concerns about ethical or regulatory requirements being met?

Reviewer #1: The work aims to investigate the phlebotomine fauna in two departments of the Colombian Amazon through an integrative approach on morphological and molecular taxonomy and the interaction of the species between parasite (Leishmania) and hosts, through the females' food source.

For the investigation, two campaigns were carried out, one in each state of Amazonas and Caqueta, with two locations sampled in the first and four in the second, with three days of collections using different collection techniques. The sampling was therefore very limited and shows a momentary snapshot, but it was possible to obtain relevant information about possible Leishmania vectors and hosts.

The statistical tests were suitable for analyzing the diversity and abundance of species in the area and support the conclusions.

The authors report the licenses obtained from the competent bodies for the development of field research.

Reviewer #2: (No Response)

Reviewer #3: the answer to all questions is yes.

**Results**

-Does the analysis presented match the analysis plan?

-Are the results clearly and completely presented?

-Are the figures (Tables, Images) of sufficient quality for clarity?

Reviewer #1: Does the analysis presented match the analysis plan? Yes

The results were clearly presented, and well illustrated, and was possible. This made it possible to question the identification of three species of sand flies two of which with relevance for the results: Trichophoromyia auraensis, the most abundant species, according to Fig. 8-B, which represents the male genitalia of Th. auraensis, seems to me to be Trichophoromyia cellulana. It is possible that Fig. 8. Neighbor-joining gene tree of sand fly COI barcode sequences, with emphasis on Trichophoromyia reflects problems with the identification of Th. auraensis. Therefore, the analysis of the results seems to me to be compromised. Furthermore, I do not know to what extent the analysis of the results would be compromised. It is possible that Th. auraensis does indeed exist together with Th. cellulana. In this case, I believe that morphologically the females are not distinguished by their spermathecae. Therefore, the results obtained would be related to the genus and not to the species.

The other taxa that seem to have identification problems are from the genus Psychodopygus: Figure 5-C, which represents the spermathecae of Ps. amazonensis, appears to be from Ps. davisi. The photo of Psychodopygus paraensis spermathecae presented in Figure 5G, is very similar to those of Ps. ayrozai. I therefore recommend checking the identification of these species not only by the spermathecae, but also by observing the characteristics of the females' cibarium. The observation of thorax of the specimens to visualize the coloration of the thoracic sclerites, which would certainly help in distinguishing the species of Psychodopygus, but as they were used for molecular tests, this is not possible.

The quality of the figures is good, but Table 1 is too wide and I could not see it in its entirety. I was able to deduce its contents from the supplementary Table 1.

Reviewer #2: (No Response)

Reviewer #3: the answer to all questions is yes.

**Conclusions**

-Are the conclusions supported by the data presented?

-Are the limitations of analysis clearly described?

-Do the authors discuss how these data can be helpful to advance our understanding of the topic under study?

-Is public health relevance addressed?

Reviewer #1: The conclusions are well supported by the data presented, but most likely they are not all correct in light of the above.

Some limitations were commented on in light of discordant results from other research, but not sufficiently explored.

The authors discuss the relevance of the analyses performed in relation to integrative analysis to assist in the determination of possible vectors. However, since the species level identification of the parasite was not performed, it was not possible to make any association between potential vectors and their respective agents.

The leishmaniases and their epidemiological components regarding population at risk o infection, agentes, possible vectors and hostes were addressed.

Reviewer #2: (No Response)

Reviewer #3: the answer to all questions is yes.

**Editorial and Data Presentation Modifications?**

Reviewer #1: none

Reviewer #2: (No Response)

Reviewer #3: (No Response)

**Summary and General Comments**

Reviewer #1: In addition to the previous comments, it was not clear how many CDC traps were installed per location. • If Shannon's traps were installed every day, how were the most productive points chosen on the first days of the campaign?

The way of citing the species names can confuse readers. For example, Trichophoromyia (Ty.) auraensis. According to the rules of zoological nomenclature, citing a name in parentheses between the name of the genus and the specific epithet (species) indicates that it is a subgenus. Therefore, I suggest citing only the genus without abbreviation, for example, Trichophoromyia auraensis. When another species of the same genus appears, then use the abbreviated name of the genus, for example Th. howardi, and so on.

The names of the authors of the genus or subgenus should not be placed in parentheses.

Other small suggestions for corrections can be found in the attached manuscript.

Reviewer #2: A well written manuscript that describes an important study of the presence of different species of sand flies known as vectors and also some that are not considered vectors infected with Leishmania sp. The study provides information on the sand fly fauna of Colombian with updates of new records for the country, information on blood feeding sources and infection rates, which are important data on leishmaniasis in an endemic region of Colombian. It also provides new COI gene sequences for some sand fly species.

Reviewer #3: The manuscript presents integrated and relevant research that combines morphological and molecular techniques to characterize the phlebotomine fauna in the regions of Amazonas and Caquetá (Colombia), with a focus on detecting DNA from blood sources and Leishmania. The topic is highly relevant to public health, especially in the context of neglected tropical diseases, and is in line with the scope of the journal. The methodology is robust, and the data is of scientific interest. The data is robust, the topic is current and highly relevant to the epidemiology of leishmaniasis from a “One Health” perspective. The scientific contribution is clear and original. However, some sections require more clarity and detail, especially in the introduction, methodology and discussion of results.

PLOS authors have the option to publish the peer review history of their article (what does this mean? ). If published, this will include your full peer review and any attached files.

**Do you want your identity to be public for this peer review?** For information about this choice, including consent withdrawal, please see our Privacy Policy .

Reviewer #1: **Yes: ** Eunice Aparecida Bianchi Galati

Reviewer #2: No

Reviewer #3: No

**Figure resubmission:****Reproducibility:** To enhance the reproducibility of your results, we recommend that authors of applicable studies deposit laboratory protocols in protocols.io, where a protocol can be assigned its own identifier (DOI) such that it can be cited independently in the future. Additionally, PLOS ONE offers an option to publish peer-reviewed clinical study protocols. Read more information on sharing protocols at https://plos.org/protocols?utm_medium=editorial-email&utm_source=authorletters&utm_campaign=protocols

---

## [Decision Letter · Decision Letter 1]

5 Aug 2025

Dear Dr. Moreno-Herrera,

We are pleased to inform you that your manuscript 'Taxonomic, molecular and ecological approach reveals high diversity of vector sand flies, varied blood source supply and a high detection rate of Leishmania DNA in Colombian Amazon region' has been provisionally accepted for publication in PLOS Neglected Tropical Diseases.

Best regards,

Nisha Singh, Ph.D.

Academic Editor

Nigel Beebe

Section Editor

Shaden Kamhawi

co-Editor-in-Chief

Paul Brindley

co-Editor-in-Chief

Manuscript may be accepted in it's current form for publication.

Reviewer's Responses to Questions

**Key Review Criteria Required for Acceptance?**

**Methods**

-Are the objectives of the study clearly articulated with a clear testable hypothesis stated?

-Is the study design appropriate to address the stated objectives?

-Is the population clearly described and appropriate for the hypothesis being tested?

-Is the sample size sufficient to ensure adequate power to address the hypothesis being tested?

-Were correct statistical analysis used to support conclusions?

-Are there concerns about ethical or regulatory requirements being met?

Reviewer #1: This is a study that investigated the sand fly fauna in two departments of Colombia with typical Amazonian characteristics. It sought to integrate taxonomic information on sand fly species, blood ingested by females, and Leishmania DNA detection, with the aim of generating knowledge about the transmission dynamics of leishmaniasis. Based on the hypothesis that the adopted approach would generalize knowledge about this transmission dynamic, the objectives are well aligned with it

The sampling effort, limited to two campaigns, was sufficient to generate information about the transmission dynamics, but clearly still requires further investigation in subsequent studies.

The statistical tests were sufficient to provide power to the hypothesis.

Regulatory aspects for the collections of the insects were detailed.

**Results**

-Does the analysis presented match the analysis plan?

-Are the results clearly and completely presented?

-Are the figures (Tables, Images) of sufficient quality for clarity?

Reviewer #1: Yes, the analysis presented in manuscipt mach the analysis plan.

Some results regarding the taxonomy presented that I questioned in the previous version have been adequately corrected. The lack of identification of Leishmania species has been explained. Ideally, specific identification would be ideal, but in any case, progress has been made in understanding the presence of leishmaniasis agents in the area and possible vectors.

The figures and tables are adequate and of good quality.

**Conclusions**

-Are the conclusions supported by the data presented?

-Are the limitations of analysis clearly described?

-Do the authors discuss how these data can be helpful to advance our understanding of the topic under study?

-Is public health relevance addressed?

Reviewer #1: The conclusions are clearly supported by the data presented.

The limitations were discussed regarding the use of a single marker to distinguish taxa from one another, which made it impossible to separate some taxa and the impossibility of identifying Leishmania species.

The study's relevance to public health was emphatically emphasized.

**Editorial and Data Presentation Modifications?**

Reviewer #1: Manuscript accepted as submitted.

**Summary and General Comments**

Reviewer #1: The authors made most of the corrections and modifications I suggested. They justified one change they considered unnecessary, and I accept the justification.

PLOS authors have the option to publish the peer review history of their article (what does this mean? ). If published, this will include your full peer review and any attached files.

**Do you want your identity to be public for this peer review?** For information about this choice, including consent withdrawal, please see our Privacy Policy .

Reviewer #1: **Yes: ** Eunice Aparecida Bianchi Galati

---

## [Editor Report · Acceptance letter]

Dear Dr. Moreno-Herrera,

We are delighted to inform you that your manuscript, "Taxonomic, molecular and ecological approach reveals high diversity of vector sand flies, varied blood source supply and a high detection rate of Leishmania DNA in Colombian Amazon region," has been formally accepted for publication in PLOS Neglected Tropical Diseases.

Best regards,

Shaden Kamhawi

co-Editor-in-Chief

Paul Brindley

co-Editor-in-Chief
